

# How well do stratospheric reanalyses reproduce high-resolution satellite temperature measurements?

Corwin J. Wright and Neil P. Hindley

Centre for Space, Atmospheric and Oceanic Science, University of Bath, Bath, UK

*Correspondence to:* Corwin J. Wright (c.wright@bath.ac.uk)

**Abstract.** Atmospheric reanalyses are data-assimilating weather models which are widely used as proxies for the true state of the atmosphere in the recent past, particularly for the stratosphere, where historical observations are sparse. But how realistic are these stratospheric reanalyses? Here, we resample stratospheric temperature data from six modern reanalyses (CFSR, ERA-5, ERA-Interim, JRA-55, JRA-55C and MERRA-2) to produce synthetic satellite observations, which we directly compare to retrieved temperatures from the COSMIC, HIRDLS and SABER instruments and to brightness temperatures from the AIRS instrument for the ten-year period 2003-2012. We explicitly sample standard public-release products in order to best assess their suitability for typical use cases. We find that all-time all-latitude correlations between limb sounder observations and synthetic observations from full-input reanalyses are 0.97–0.99 at 30 km altitude, falling to 0.84–0.94 at 50 km. The highest correlations are seen at high latitudes and the lowest in the sub-tropics, but root-mean-square (RMS) differences are highest (10 K or greater) in high-latitude winter. At all latitudes, differences increase with increasing height. High-altitude differences become especially large during disrupted periods such as the post-sudden stratospheric warming recovery phase, where zonal-mean differences can be as high as 18 K between different datasets. We further show that, for the current generation of reanalysis products, a full-3D sampling approach is always required to produce realistic synthetic AIRS observations, but is almost never required to produce realistic synthetic HIRDLS observations. For synthetic SABER and COSMIC observations full-3D sampling is required in equatorial regions and regions of high gravity-wave activity but not otherwise. Finally, we use cluster-analyses to show that full-input reanalyses are more tightly correlated with each other than with observations, even observations which they assimilate. This may suggest that these reanalyses are over-tuned to match their comparators. If so, this could have significant implications for future reanalysis development.

## 1 Introduction

One of the most important tools in the atmospheric sciences is the reanalysis. These are are weather models which assimilate observations from the historical record, using a fixed analysis scheme to consistently simulate an extended period, typically decades. Particularly in the stratosphere, where measurements remain comparatively sparse even in the modern era, they are widely used as a proxy to the true atmospheric state for purposes as diverse as tuning models and understanding geophysical processes. This is because they provide a spatially- and temporally-uniform estimate of the atmospheric state, with a wide range of variables such as temperature, wind and vorticity available in a standard format.



For reasons of numerical stability and dynamical balance, reanalyses must always favour the model state where significant conflicts exist. Thus, although they are observation-driven in a much more direct way than standard atmospheric models, biases remain between the reanalysis and observed states. Given the extremely wide use of reanalysis data, it is important to quantify these biases.

This quantification is typically carried out in two main ways. The first of these is to implement a forward-model software chain, often referred to as an 'observation operator', in the reanalysis system (e.g. Dee et al., 2011). This method is commonly used for quality control and validation inside reanalysis centres, but requires direct access to the reanalysis system rather than the final output product. A second approach, common when developing new satellite instruments but also applicable to validating data from existing ones, is to run a high-resolution variant of the reanalysis system, often known as a 'Nature Run'

(e.g. Nolan et al., 2013; Holt et al., 2016). The Nature Run is then combined with an observing system simulation, which properly samples the model as the satellite, including the full data retrieval chain. Both of these methods inherently customise the reanalysis output rather than use the standard output formats common in non-observational research. The latter method also requires a reimplementation of the full satellite data retrieval.

Here, we wish to assess standard public-release reanalysis products, in order to quantify the validity of their *de facto* broad

use as an observational substitute. This rules out the above approaches. We instead develop and apply a simple comparison method based on (1) oversampling the reanalysis data grid, and (2) re-weighting the oversampled fields in a close approximation to how the instruments weight observations of the true atmosphere. This allows us to produce weighted composite reanalysis samples of equivalent form to the satellite measurements. In this study, we will refer to these samples as 'synthetic measurements'.

We specifically compare synthetic temperature measurements produced from six reanalyses (CFSR, ERA-Interim, ERA-5, JRA-55, JRA-55C and MERRA-2) to three observational datasets (COSMIC, HIRDLS and SABER). Low-level bending angle data from COSMIC are assimilated by all six reanalyses, whilst HIRDLS and SABER are not assimilated. We compare the reanalyses to all temperature measurements from these three instruments for the ten-year period 2003-2012. We also produce synthetic temperature measurements using the sampling characteristics of AIRS brightness temperature measurements, and

use these to demonstrate the importance of using a full sampling approach for comparisons to instruments of this type. Our method is generalisable to other atmospheric models, instruments and variables.

Sections 2 and 3 describe the observational and reanalysis datasets used. Section 4 describes the sampling method used, including an explanation of the motivation underlying our approach. Section 5 then assesses the performance of our sampling method relative to the simpler approach of interpolating the reanalysis to the measurement location. Sections 6 – 9 then discuss

differences between the synthetic and true observations as a function of time and geographic location. Finally, Section 10 analyses the bulk-scale differences between each dataset, before we summarise and draw conclusions in Section 11.



## 2 Data

### 2.1 Satellite Instruments

We consider observed data from and the scanning patterns of four satellite instruments. These are the Advanced InfraRed Sounder (AIRS), the Constellation Observing System for Mesosphere, Ionosphere and Climate (COSMIC), the High Resolu-
tion Dynamics Limb Sounder (HIRDLS), and Sounding of the Atmosphere By Broadband Emission Radiometry (SABER).

Of these, two (HIRDLS and SABER) are limb-sounding radiometers, one (AIRS) is a hyperspectral nadir sounder, and one (COSMIC) is a multi-satellite constellation which uses GPS radio occultation to infer atmospheric parameters. GPS radio occultation is arguably a special case of limb sounding, and thus throughout this paper we will refer to COSMIC, HIRDLS and SABER collectively as limb sounders and to AIRS as a nadir sounder.

### 2.2 AIRS

The Advanced Infrared Sounder (AIRS) is an instrument on NASA's Aqua satellite, launched in May 2002 (Aumann et al., 2003). Part of NASA's A-Train satellite constellation, Aqua has a 98-minute sun-synchronous polar orbit, with an equator-crossing local solar time of 1.30 pm in the ascending node. AIRS has 2378 spectral channels, which provide a continuous swath of radiance measurements. Its cross-track footprint width averages ∼20 km, varying from 13.5 km at the centre of the
instrument track to 40 km at the edges across 90 parallel tracks (Olsen et al., 2007). This across-track variation is due to the interaction of the rotated scanning volume with atmospheric density in the vertical plane.

We use brightness temperatures derived from AIRS Level 1 (version 5) radiance data in the 667.67 cm$^{-1}$ channel, which is centred at ∼42 km altitude. We refer to these data subsequently as AIRS-L1. These radiance data are available at considerably higher horizontal resolution than the standard AIRS Level 2 temperature product (Hoffmann and Alexander, 2009), and are
consequently useful for studying small-scale phenomena such as gravity waves. This is because they preserves wave features in the vertical, which the methods used to optimise the standard AIRS Level 2 product suppress (Alexander and Barnet, 2007). The noise-equivalent $\Delta$T for measurements at these altitudes is ∼0.7 K (Hoffmann et al., 2014).

Accordingly, our sampled measurements, while well-suited to measuring small-scale perturbations to temperature, are not useful for absolute temperature comparisons. In this paper, we therefore include AIRS-L1 in comparisons between two
reanalysis-derived properties, but not otherwise. Future work will compare gravity wave measurements derived from perturbations to these synthetic data to real AIRS measurements.

### 2.3 COSMIC

The Constellation Observing System for Meteorology, Ionosphere and Climate (COSMIC) is a constellation of six microsatellites launched in 2006, of which three remain active at time of writing. Each satellite carries a radio occultation receiver, which
is used to intercept GPS signals transmitted through the atmosphere. The phase delay in these signals allows the bending angle





of the signal path through the atmosphere to be computed, which can then be analysed to produce profiles of stratospheric dry temperature (Anthes et al., 2008).

COSMIC measurements are distributed pseudo-randomly across the globe. Around 1000–2000 such profiles are measured globally per day, with the number declining as satellites have aged. Temperature soundings typically cover the 5–50 km altitude

range. The vertical resolution in the stratosphere, estimated based on the size of the signal Fresnel zone, is ∼1.5 km (Kursinski et al., 1997), with a precision of ∼0.5 K (Anthes et al., 2008). Horizontal resolution is also ∼1.5 km in the across-line-of-sight direction, but is approximately 270 km in the along-line-of-sight direction due to path integration. We use COSMIC version 2013 data, which currently extends from the beginning of data availability in June 2006 until the end of April 2014.

## 2.4 HIRDLS

The High Resolution Dynamics Limb Sounder (HIRDLS) is a 21-channel radiometer on NASA's Aura satellite. Designed to measure stratospheric dynamics and chemistry at high vertical resolution, HIRDLS is also part of NASA's A-Train satellite constellation. Thus, Aura also has a sun-synchronous orbit, with an equator-crossing time a few minutes after Aqua.

Due to an optical blockage discovered shortly after launch, HIRDLS data products have required significant corrections to be usable for scientific purposes (Gille et al., 2008). Following these corrections, validation has shown that temperature

products are comparable to those from other data sources, with an estimated precision ∼0.5 K and a vertical resolution of 1 km throughout the stratosphere, falling to 2 km in the mesosphere (Gille et al., 2013). Horizontal resolution for individual measurements is ∼200 km in the along-line-of-sight direction and ∼20 km in the across-line-of-sight direction. Due to the nature of the blockage, the instrument line of sight is directed 47° off-track in the opposite direction to satellite travel. Data are available from early 2005 until early 2008, when a problem with the optical chopper terminated data collection.

## 2.5 SABER

Sounding of the Atmosphere by Broadband Emission Radiometry (SABER) is a 10-channel limb-sounding infrared radiometer aboard the TIMED satellite. SABER provides ∼2200 profiles globally per day, with a vertical resolution of ∼2 km and an along-track profile spacing alternating between 200 km and 550 km, and a line of sight lying 90° off-track. Kinetic temperature profiles cover the 15 km-120 km altitude range, with a precision of ∼0.8 K in the stratosphere (Remsberg et al., 2008).

Horizontal resolution for individual measurements is ∼300 km in the along-line-of-sight direction and ∼50 km in the across-line-of-sight direction. Coverage is continuous between 50°S and °N throughout the year, extending to either 80°S or 80°N on an alternating 60-day yaw cycle.

## 2.6 Relative Sensitivity

Figures 1 and 2 show the approximate sensitivity of each instrument to atmospheric temperature (or 667.67 cm$^{-1}$ radiances for

AIRS). This is shown in Figure 1 as geolocated 3D volumes which are defined to contain 99% of the total estimated weight associated with each measurement. This figure consequently gives an indication of the relative measurement volumes and



spacing associated with the standard instrument scanning patterns. Figure 2, meanwhile, shows the sensitivity associated with a single measurement (specifically at nadir in the case of AIRS and below 60 km altitude in the case of HIRDLS), plotted as 2D cuts through a sensing volume centred at the origin. These show that the majority of the signal in all cases comes from near the measurement centre (Section 5 quantifies this further), but also that the total integrated signal in each case can represent

quite different total volumes even for perfectly-overlapping measurements.

Note that, in our analysis, we linearly downsample all three limb sounders to their estimated vertical resolution (i.e. 1 km for HIRDLS, 1 km for COSMIC, and 2 km for SABER), rather than use their original measurement locations (typically spaced by hundreds of metres). This reduces the number of samples significantly, providing a large reduction to our overall computational requirements, but ensures that we still recover a full vertical profile of independent measurements. A full-resolution approach

was tested, and was found not to affect results significantly.

## 3   Reanalyses

In this study, we produce synthetic measurements from six reanalyses using the sampling patterns and sensitivity functions of the above four instruments. These reanalyses are CFSR, ERA-Interim, ERA-5, JRA-55, JRA-55C and MERRA-2. Each of them is widely used in the scientific community for a variety of purposes. With the exception of JRA-55C, they are all generated

using full-input reanalysis systems, i.e. they assimilate both surface and upper-atmospheric conventional and satellite data. JRA-55C assimilates a full suite of surface and near-surface observations, but no upper-atmospheric satellite data, and thus provides a useful test of how well the model physics transmits information to the middle and upper atmosphere: in particular, this is useful for assessing how well-constrained the other reanalyses can be considered to be before the satellite era.

Table 1 and Figure 3 describe the key relevant properties to our study of each reanalysis, including the horizontal and

vertical resolutions of the products used and whether they assimilate the instruments we consider. In general, COSMIC is assimilated by all reanalyses, AIRS by most, and SABER and HIRDLS by none. Beyond these details, the S-RIP special issue of Atmospheric Chemistry and Physics (Fujiwara et al., 2017, and references therein) provides an extremely detailed summary of the key features of each reanalysis, and thus we omit a general description for brevity. Relevant details will be referenced directly in the text where appropriate. Further details on each reanalysis system can be found in Saha et al. (2010) [for CFSR],

Dee et al. (2011) [ERA-Interim], Kobayashi et al. (2014) and Kobayashi et al. (2015) [JRA-55 and JRA-55C], and Gelaro et al. (2017) [MERRA-2]; no equivalent reference exists for ERA5 at time of writing.

## 4   Sampling Method

### 4.1   Concept

Figure 4(a) illustrates our sampling approach. The blue circles show a 100 km resolution grid, intended for illustrative purposes.

For comparison, our lowest resolution analysis, ERA-Interim, has a horizontal resolution ~84 km at the Equator. Thus, while this grid is slightly coarser than reality to clarify our explanation, it is not excessively so.





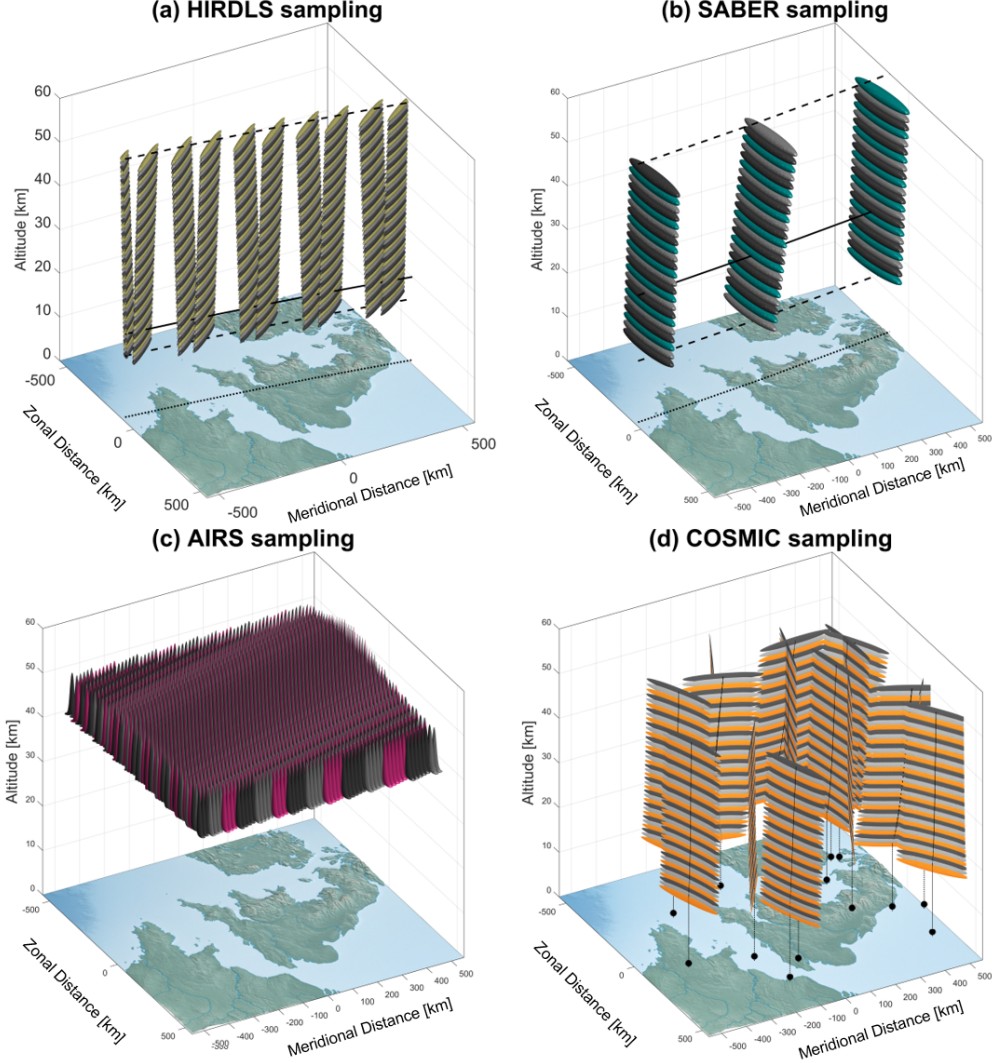

**Figure 1.** Approximate averaging volumes for (a) HIRDLS (b) SABER (c) AIRS-L1 (d) COSMIC, shown over Northwestern Europe for scale. Solid volumes show the regions which contribute the largest 99% of the signal for each measurement. For (a) and (b), solid lines show the scan track at the reference altitude for that instrument (20 km for HIRDLS, 30 km for SABER), dashed lines show the track at 15 km and 60 km altitude, and dotted lines show the track at 0 km altitude. For (d), black lines indicate centre of measurement volume for each profile at each height, black circles indicate the surface geographic location corresponding to the profile centre at 15 km altitude, and the dotted lines join these. Alternating colours are used for contrast between individual averaging volumes, and do not indicate any technical difference.

The shaded cyan oval, meanwhile, shows a horizontal sensitivity function typical of SABER. Any real SABER temperature measurement will be a single number representing the weighed average of temperature across this ovoid (i.e. including its ver-





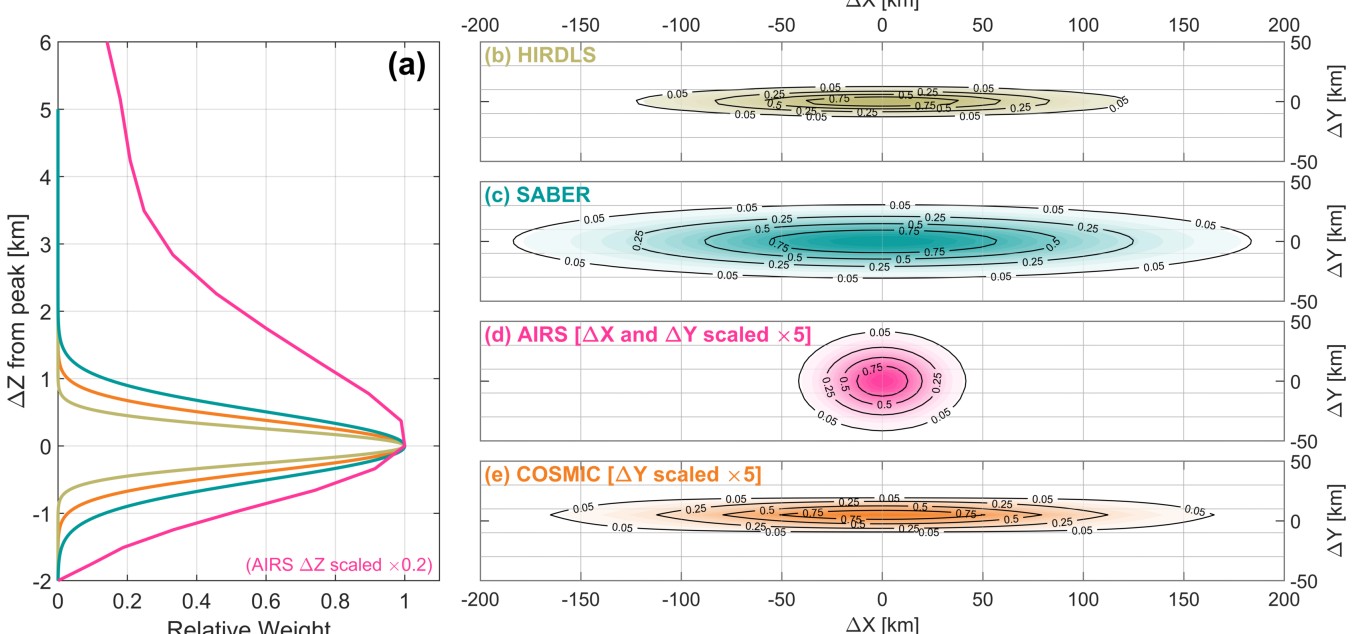

**Figure 2.** Sensitivity functions for each instrument in (a) the vertical direction and (b-e) the horizontal plane. $\Delta X$ is the major axis and $\Delta Y$ the minor axis in the horizontal plane, and $\Delta Z$ the vertical axis, with 0 in each direction defined as the peak sensitivity. Sensitivities are normalised to a peak value of 1.

**Table 1.** Key properties of the six reanalyses used in this study. 'Relevant period' refers to the time period covered by both the reanalysis and at least one of the instruments we use in this study; in all cases, we terminate data analysis at the end of 2012 if it continues past that date.

| Reanalysis | Relevant Period | Resolution | | | | Does It Assimilate...? | | | |
|---|---|---|---|---|---|---|---|---|---|
| | (yyyy/mm) | Time (h) | Lon (°) | Lat (°) | Vertical | AIRS | COSMIC | HIRDLS | SABER |
| CFSR | 2003/01 - 2011/03 | 6 | 0.5 | 0.5 | 64 levels, top 0.625 hPa | Y | Y | N | N |
| ERA-Interim | 2003/01 - 2012/12 | 6 | 0.75 | 0.75 | 60 levels, top 0.1 hPa | Y | Y | N | N |
| ERA-5 | 2010/01 - 2012/12 | 1 | 0.3 | 0.3 | 137 levels, top 0.01 hPa | Y | Y | N | N |
| JRA-55 | 2003/01 - 2012/12 | 6 | 0.56 | 0.56 | 60 levels, top 0.1 hPa | N | Y | N | N |
| JRA-55C | 2003/01 - 2012/12 | 6 | 0.56 | 0.56 | 60 levels, top 0.1 hPa | N | Y | N | N |
| MERRA-2 | 2003/01 - 2012/12 | 3 | 0.625 | 0.5 | 72 levels, top 0.01 hPa | Y | Y | N | N |

tical extent, not shown). There is an obvious spatial mismatch between the ovoid and the reanalysis gridpoint locations, which makes direct comparison challenging. The same fundamental concept holds true for any other remote-sensing measurement, which will have a definite weighted volume of some form which does not usually correspond to a comparator model.

5    If we are working with standard reanalysis products, as is the case here, this mismatch can be overcome in three main ways. The first such approach is simply to interpolate the model to the centre of the measurement volume, i.e. (0,0) on these axes.



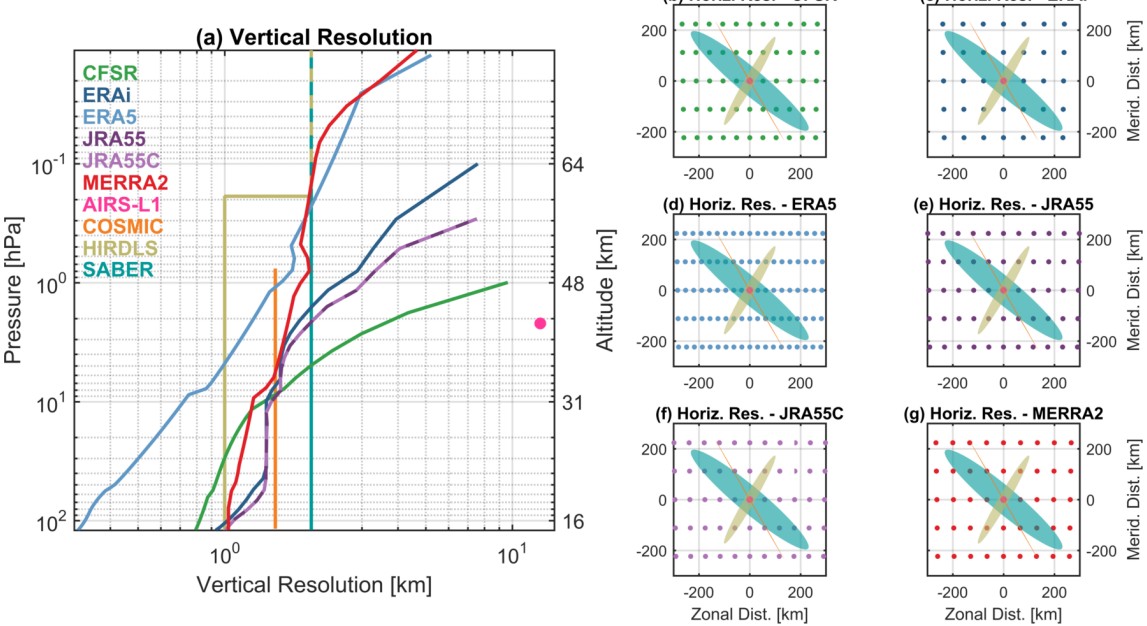

**Figure 3.** (a) Vertical and (b-g) horizontal resolution of each dataset used. For each of panels (b)–(g), the reanalysis grid is shown at an arbitrary centre latitude of 70°N to demonstrate the curvature of the spatial grid at high latitudes, and overlaid with the approximate horizontal weighting volume for each instrument, rotated to different angle to minimise overlap.

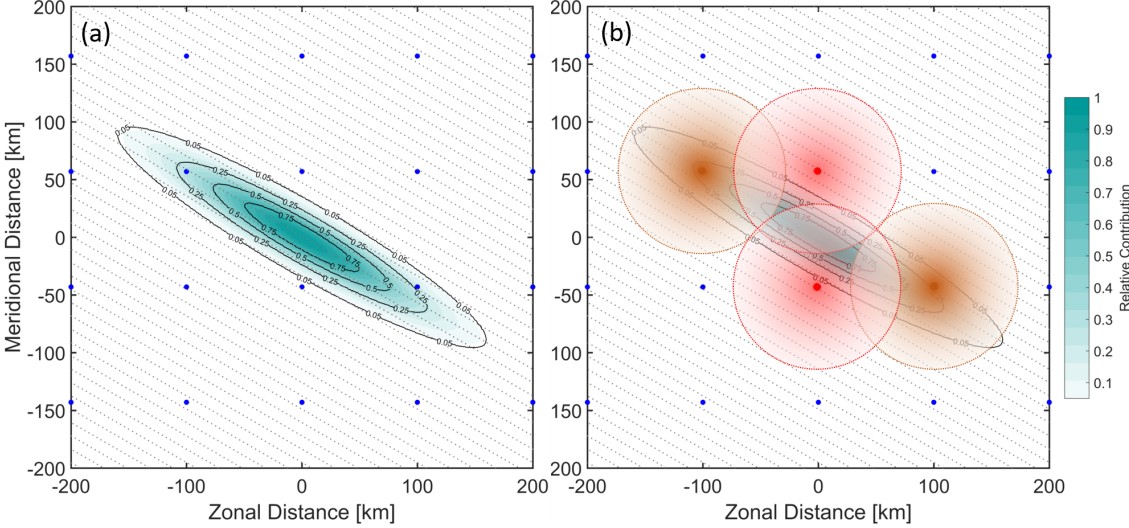

**Figure 4.** 2D schematic illustrating the motivation underlying our sampling approach. See text for further details.





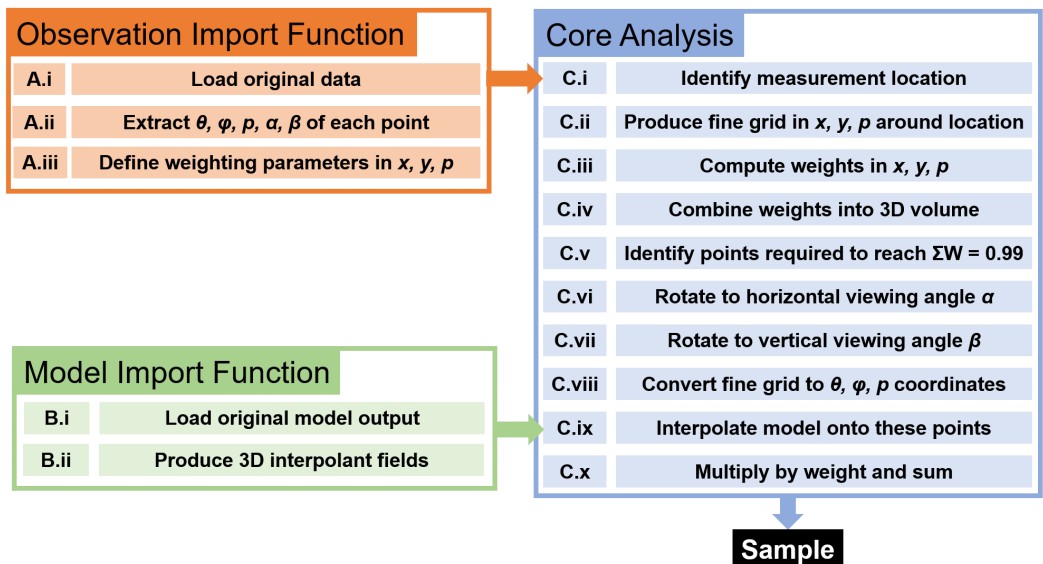

**Figure 5.** Schematic outline of the sampling process. See text for further details.

This is a viable approach for many uses, if not most, since the majority of the sensing weight is here. Section 5 assesses the accuracy of this relative to the approach we do adopt.

A second approach is to produce a weighted sum of the model gridpoints inside the sensing region. This can work well for
5    coarse measurements, for example those of the SSU and AMSU sounders, but breaks down where the instrument resolution is comparable to the model grid. Indeed, for high-resolution instruments this can produce a less representative result than simple interpolation. This is illustrated by Figure 4(b). Here, brown circles indicate points which would contribute to a SABER-equivalent reanalysis value computed using only points inside the measurement volume. Red circles indicate model points which would not contribute to this sum, but which should have a larger contribution than the brown points. This method can
10   thus discard highly-relevant local data, while overweighting more distant points. While in practice for SABER, more points would lie inside the ovoid at the resolution of a real reanalysis, this would not be the case for HIRDLS and COSMIC, where it is quite possible no points will lie inside a given volume.

A third approach, and the one we use in this study, is to first oversample the model data onto a much finer grid (grey dots). By weighting these fine grid points appropriately and then summing, we can combine all relevant information and ameliorate the problems of the second approach significantly, while still providing an improvement in accuracy over simple interpolation.



## 4.2 Method

Figure 5 outlines our implementation of this sampling method. The method can be broadly divided into three parts: an 'Observation Import Function' ('OIF'), a 'Model Import Function' ('MIF'), and the 'Core Analysis' ('Core'). These three parts are
described individually below.

This scheme provides significant flexibility of analysis. Both the OIF and MIF can be easily substituted, allowing the analysis of a wide range of datasets. In this study, we use four OIFs and six MIFs, representing the instruments and reanalyses described above, and plan to develop more for future model/observation comparison work. Further, no inherent assumptions are made as to the nature of the data being analysed, provided a suitable sensitivity function is provided. For example, while in this study
we consider only temperature data, it is perfectly possible to instead sample reanalysis wind speeds or chemical distribution fields for comparison to observations.

### 4.2.1 Observation Import Function

The OIF reads in the observational data in its original format, and outputs observational parameters needed for the later stages of the analysis. These parameters can be divided into two groups of parameters: (a) geolocation and (b) sensing volume. All
values are computed and stored at the individual measurement level.

The geolocation parameters we use are the (i) latitude $\phi$ (ii) longitude $\theta$ and (iii) pressure level $p$ of the centre of each measurement, together with the (iv) horizontal $\alpha$ and (v) vertical $\beta$ viewing angle at which the measurement was taken. For all observations used in this study, the latitudes, longitudes and pressures are provided in the original data files used and are simply duplicated into the appropriate format for feeding into the Core.

Horizontal viewing angles are defined as clockwise from geographic north. For COSMIC, the original files provide this information, which is simply duplicated. For AIRS-L1, SABER and HIRDLS, we combine the known scanning geometries of these instruments (47° off-reverse-track for HIRDLS, 90° off-track for SABER, directly below for AIRS) with the time-varying measurement latitudes and longitudes to compute horizontal viewing angles for each data point.

Vertical viewing angles are defined as anticlockwise from instrument nadir. These are defined as zero for COSMIC, HIRDLS
and SABER, since all three instruments observe in the Earth's limb from a considerable distance and are thus near-horizontal at any given height in the vertical plane on the scales under consideration. For AIRS-L1, the vertical viewing angle is defined as the angle of the individual cross-track row from the primary travel axis, which ranges from -49.5° to +49.5°.

The sensing volume parameters are specified based on prior knowledge about the instrumentation used. For the limb sounders, we approximate the sensing volumes using 1D Gaussians. Thus, the OIF specifies for each point simply the standard
deviation of an appropriately-sized Gaussian function in each dimension, which are combined in the core to produce a 3D sensitivity weighting volume. For AIRS, we use the same approximation in the horizontal plane, but use a vertical function of an appropriate form for AIRS sensitivity (Figure 2), which is specified precisely by the OIF.





### 4.2.2  Model Import Function

The MIF imports the model data, and uses it to produce a linear interpolant field for later use. Specifically, for each model time step, the MIF imports the data from the original format, and reformats it to a binary-search-suitable fast-lookup format.

This allows us to rapidly interpolate the model data field to any geographic location without additional data reformatting, a capability which is exploited in the core analysis below.

An important choice at this stage is whether to interpolate the data in time as well as space. For the reanalyses used, the data fields are instantaneous, i.e. a snapshot of the model at a particular time as opposed to an average over the time surrounding the time step. Thus, small-scale features, such as internal gravity waves and chemical laminae, are preserved in the fields.

If we wish to preserve these features in our sampling, it would be unwise to interpolate in time. However, we find that if we do not interpolate in time, the difference between the sampled and observed temperature fields depends strongly on the time difference between the model time and observation time, due to effects such as the diurnal cycle. In extreme cases, this can create additional deviations as large as 10 K. Accordingly, throughout this study, we linearly interpolate our input data to the sampling location in time as well as space.

## 4.3  Core

The Core takes the output of the OIF and the MIF, and then resamples the model data appropriately onto the satellite measurement track to produce the required synthetic measurements. The Core analysis consists of ten distinct steps (Figure 5); of these, nine are scientifically motivated, while one (C.v) exists to reduce processing time.

### 4.3.1  Produce Fine Grid (steps C.i to C.ii)

We first identify the measurement centre location supplied by the OIF, and produce a fine spatial grid around this location. This grid is defined such that the $x$ axis lies along the major axis of the measurement and the $y$ axis lies along the minor axis, with the pressure ($z$) axis vertically aligned through the measurement centre. The limits of the fine grid are defined as the three standard deviation point for weights specified as Gaussians, i.e. all cases in this study except for AIRS-L1 in the vertical. For AIRS-L1 we terminate the grid half a decade of pressure above and below where the weight falls to 2% of its peak value.

This coordinate system ensures that, regardless of the density of the fine points, at least one (the centre) point will always be contained in the measurement volume. However, in practice, significantly more points are used. Specifically, we use the fine grid spacings shown in Table 2, chosen using the sensitivity tests described in Appendix A.

### 4.3.2  Produce and rotate weighting volume (steps C.iii to C.vii)

We next define the fractional contribution of each point on the fine grid to the final synthetic measurement. To do this, we
first compute the appropriate Gaussian in each dimension separately (except AIRS-L1, where we use the specified vertical function), and then multiply these 1D functions together appropriately to produce a 3D weighting volume.



**Table 2.** Fine sampling point spacings used for each instrument

|         | Pressure [dec. pres.]        | Across-Track [km] | Along-Track [km] |
|---------|------------------------------|-------------------|------------------|
| AIRS-L1 | $1/20$ ($\sim$800 m)         | 3.0               | 10.0             |
| COSMIC  | $1/80$ ($\sim$200 m)         | 0.5               | 10.0             |
| HIRDLS  | $1/80$ ($\sim$200 m)         | 3.0               | 10.0             |
| SABER   | $1/80$ ($\sim$200 m)         | 2.0               | 10.0             |

Of the several thousand points typically contained in this volume, a great majority make no significant contribution to the total weighted sum. Passing these points through the subsequent analysis is thus computationally expensive for minimal benefit. Accordingly, we sequence the points in descending order by fractional contribution to the total signal, and then select the points needed to reach a cumulative total of 99% of the total weight. Tests show that this makes no discernible difference to the final result (typically differences are $<1\times10^{-4}$ K), but reduces runtime by 90%.

We then need to rotate the volume into the appropriate horizontal and vertical viewing angle. For horizontal angles, this is simply achieved by applying the necessary coordinate transform.

In the vertical, this is more complex, due to the strong dependence of the weights on atmospheric density. Thus, to rotate in the vertical, we first unscale the weights at each height using a density climatology derived from SABER observations. We then apply the appropriate rotation matrix for the necessary coordinate transform, before rescaling by density in the new height coordinates. In this study we only do this for AIRS; the curved extensions visible at the bottom of each coloured volume on the edge of the swath in Figure 1 arise due to this rotation, leading to reduced horizontal resolution at track edge consistent with how the real instrument observes the atmosphere.

### 4.3.3 Interpolate and Weight (steps C.viii to C.x)

Finally, we convert our fine grid $x$, $y$ and $p$ coordinates to $\phi$, $\theta$ and $p$ coordinates, using their geometric location relative to the centre location of the original measurement. We then interpolate the global model onto these coordinates, apply the weights, and sum to produce a single temperature value. This is our final synthetic temperature measurement, which we treat in subsequent analyses as if it were a satellite temperature measurement. The whole process then repeats independently until all measurements required have been sampled.

## 5 Full Sampling vs Single-Point Sampling

Our sampling approach is relatively computationally expensive. A much cheaper approach would be to instead simply interpolate the reanalysis temperature field to the measurement-centre locations, i.e. a single point approximation to the satellite sensing function. It is thus useful to quantify how large a difference the more complex approach makes to the final results. The full details of this comparison are shown in Appendix B, and we only outline the conclusions here.





We refer to these approaches as the 'SPA' and 'full' approaches and to the difference between the resulting temperature estimates as $\Delta T_{SPA}$. When considering bulk temperatures (as opposed to perturbations), we find that a full sampling approach is required for comparison to AIRS but remains unnecessary even with the most advanced current reanalyses for all limb

sounders considered: while $\Delta T_{SPA}$ can be comparable to instrument *precision* in some cases, it is usually much smaller than uncertainties due to instrument *accuracy*, which is affected by other factors in the retrieval process.

However, when considering smaller-scale perturbations to the data, which are important in fields such as gravity wave and turbulence studies, the size of $\Delta T_{SPA}$ relative to instrument precision and to the magnitude of the signals under investigation becomes more significant. For this type of work, we find that:

1. the full sampling approach is always justified for comparisons to AIRS-L1 data, even in the bulk temperature case, since >80% of samples in all reanalyses have a $\Delta T_{SPA} > 1$ K.

2. while we do so for the rest of this study for reasons of internal consistency[1], no current reanalysis justifies the use of full rather than SPA sampling when comparing to HIRDLS data. This is because $\Delta T_{SPA}$ is almost always (100% of the time for JRA55, JRA55C and ERA-I, 99.7% of the time for CFSR and MERRA2) smaller than the instrument precision at

HIRDLS lengthscales . This is an especially important point given the relatively large sampling time per day required (~20 minutes per day of data, compared to ~5 min/day per SABER, due mostly to the larger number of profiles) and Appendix A).

3. **in some cases**, the full sampling approach is justified for COSMIC and SABER data, particular when using high-resolution reanalyses such as ERA5. Cases where it is useful to do so include sampling near the equator and in regions

of high gravity wave activity. While the global percentage of samples with $\Delta T_{SPA}$ comparable to instrumental precision is only ~5% or less, such samples are preferentially located in these regions (Figure B2). Furthermore, the use of pairs of vertical profiles for typical GW analyses (e.g. Ern et al., 2004; Alexander et al., 2008), with each profile containing dozens of individual samples, means that a relatively small percentage of large $\Delta T_{SPA}$ samples in such a region could easily affect a significant proportion of wave estimates made using SPA-sampled reanalysis data.

The remainder of this paper will use our synthetic measurements to compare bulk temperatures between COSMIC, HIRDLS, SABER and our six reanalyses. Based on the above conclusions, we expect the difference from the results that would be obtained using a SPA approach to be extremely small, such that the majority of differences will arise due to scanning patterns and retrieval methods rather than the use of full-3D sampling. However, our results show that as reanalyses continue to improve in resolution, using a full sampling approach will become an increasingly important factor in producing accurate comparisons.

## 30  6  Relative Scatter

Figure 6 compares observed and synthetic temperatures for each of the three limb sounders at the 30 km altitude level. Results are plotted as a scatter-density plot for each instrument-reanalysis combination. All profiles at all latitudes are included in

---

[1]This is appropriate here as the data were already generated to compute the full statistics of $\Delta T_{SPA}$, and thus no additional computational time is required.





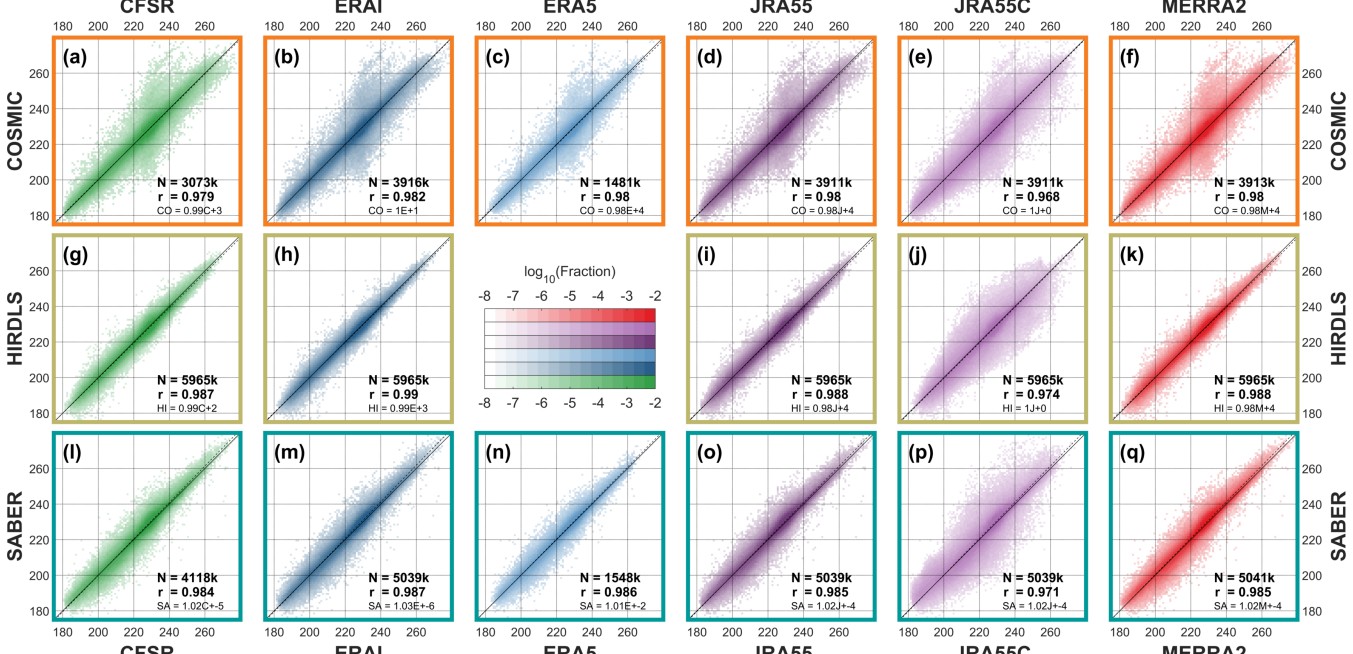

**Figure 6.** Scatter-density plots comparing each instrument (rows) to synthetic measurements generated from each reanalysis (columns) at 30 km altitude. For each panel, pixels indicates the proportion of all samples with that (vertical axis) observed and (horizontal axis) simulated temperature. The solid black line indicates a 1:1 correspondence between the datasets, and the dashed line a linear fit to the results. Text annotations show the number of contributing sample-pairs ($N$, in thousands of samples), the Pearson linear correlation coefficient of the two datasets ($r$), and the equation of the linear fit line (in the form $y = mx + c$, where $y$ is the instrument, $x$ the sampled model, $m$ the gradient and $c$ the $y$-axis intercept.)

the comparison, and thus each panel summarises at least 1.4 million like-for-like comparisons between an instrument and a reanalysis.

In all cases, agreement is, to first order, very close. No correlation coefficient is less than 0.968, with the HIRDLS-ERA-I pairing generating the highest correlation of 0.990. Fitted gradients are also extremely close, with a minimum of 0.98 and with several pairs showing a perfect linear fit, i.e. a fit gradient of 1.00 with zero offset.

JRA-55C exhibits the worst agreement with the observations, consistent with the lack of stratospheric data assimilation in this reanalysis. However, even this pairing shows a minimum correlation of 0.968 (with COSMIC) and a maximum spread about the 1:1 line ∼15 K in the most extreme case (a few dozen samples in 3.9 million). This suggests that the model physics is still doing a good job of reproducing the bulk characteristics of the observed atmosphere at the 30 km level, and thus that pre-satellite-era reanalyses are potentially scientifically useful at these altitudes.

COSMIC comparisons exhibit more scatter than HIRDLS and SABER comparisons. Since all individual comparisons are shown on the chosen colour scale, (the minimum coloured value, $10^{-7.67}$, is smaller than the smallest possible bin value), this





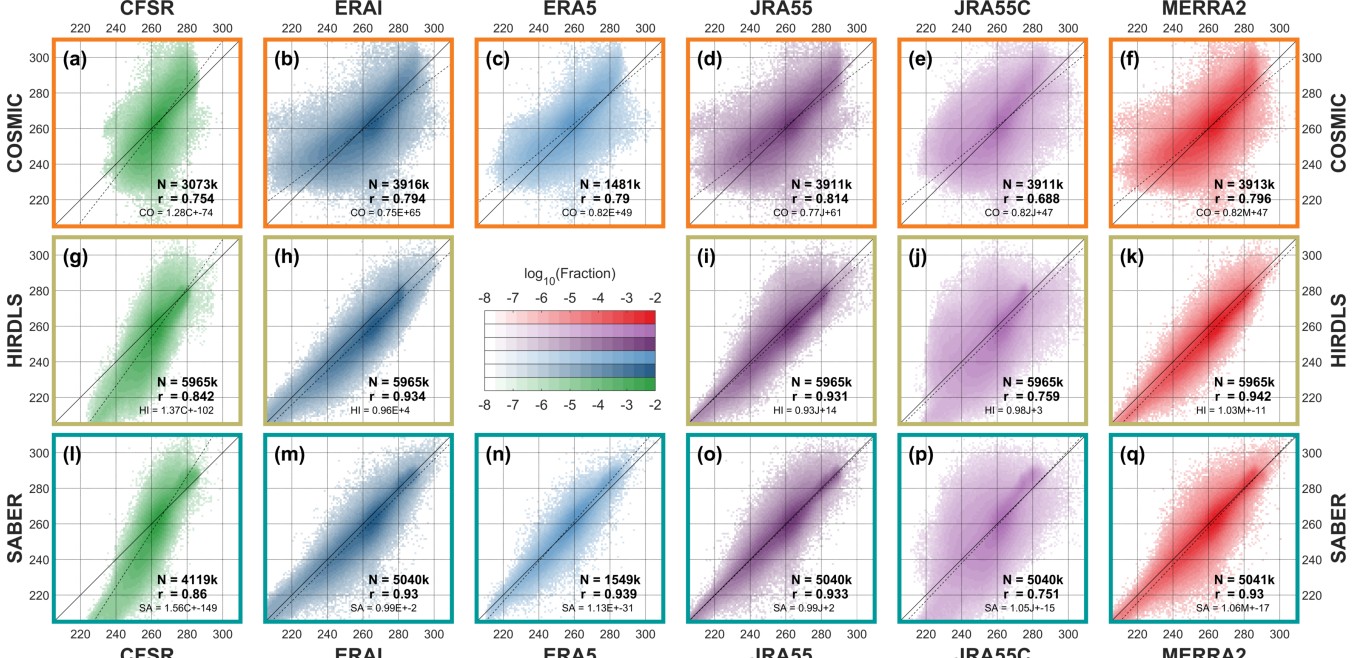

**Figure 7.** As Figure 6, but at 50 km altitude.

is not an artefact of the number of points, and in any case, the number of samples is broadly comparable to SABER (~75% as many samples). This may suggest greater noise on the COSMIC measurements. However, it may alternatively be due to their different spatial coverage. COSMIC measurements extend over the poles while SABER data terminate at either 80° or

50° depending on yaw phase - since the poles have both comparatively poor satellite coverage from the global meteorological observing constellation and are also physically complex to model due to known issues with small-scale effects such as wave forcing (e.g. Butchart et al., 2011; Garcia et al., 2017), this may lead to greater discrepancies in our comparison.

Figure 7 show equivalent results at 50 km altitude. This is more than a decade of pressure above Figure 6, and thus both simulating and retrieving atmospheric temperature is a much larger scientific challenge.

While this altitude is well within the coverage range of HIRDLS and SABER, it is very close to the upper limit of COSMIC data. Here, measurement effects can significantly affect the retrieved dry temperature, resulting in much greater retrieval noise than at lower altitudes. At low temperatures and densities, the refractive index of the atmosphere is also low, and hence the phase delays measured by the COSMIC GPS-RO receivers are very close to the instrumental noise limit imposed by thermal noise effects (Tsuda et al., 2011). This leads to a positive bias at low temperatures, since the phase delay noise is often larger

than the temperature-induced phase delay which the retrieval aims to measure. In addition to a general widening of the scatter-cloud, this also manifests itself as a characteristic 'banana' form visible in the comparisons with all six reanalyses. We also see a slight cold bias in HIRDLS temperatures of a few Kelvin, consistent with previous studies (e.g. Gille et al., 2013).





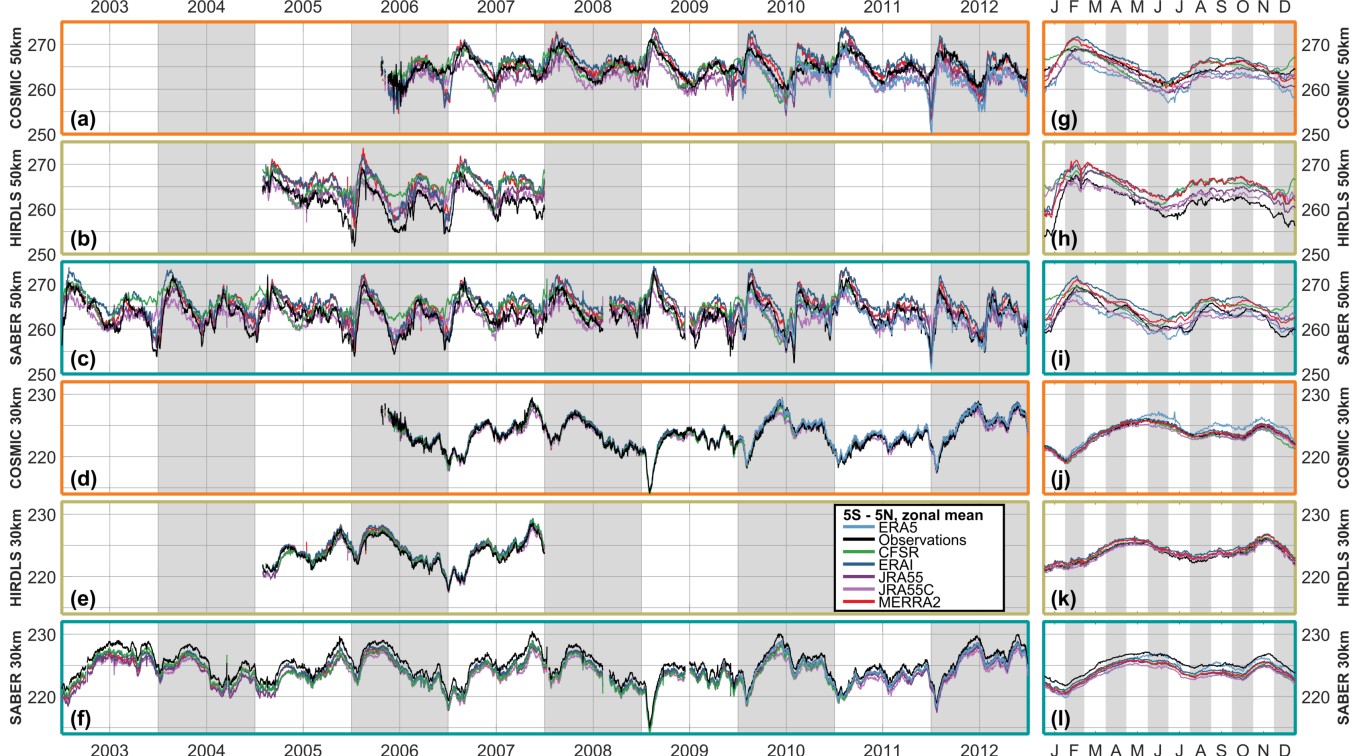

**Figure 8.** Time series of observed and sampled-reanalysis temperature, averaged over $5°S - 5°N$, at (a-c, g-i) 50 km (d-f, j-l) 30 km altitude. Panels (a-f) show time series over the period 2003-2012, and panels (g-l) the annual cycle derived from the extended time series.

Aside from these known features, some clear trends in the comparisons can be seen. Firstly, an increase in scatter for all comparisons is clearly visible. In particular for JRA-55C, correlation coefficients have dropped from 0.974 to 0.759 for HIRDLS and from 0.971 to 0.751 for SABER. This is a large drop, and shows that the model is performing much worse as we

5   get further from assimilative constraints in the lower atmosphere. Even discounting comparisons involving either COSMIC or JRA-55C, correlation coefficients now range from 0.842 – 0.942 at this height, compared to 0.984 – 0.990 at 30 km altitude, a substantive drop indicative of our poorer state of knowledge of the atmosphere at these heights.

Another noticeable effect is that, while the ERA-I, ERA5, JRA55, MERRA2 and even JRA55C comparisons with HIRDLS and SABER all have a linear trend very close to the 1:1 correspondence line, CFSR has a noticeably different gradient -

10   observed cold temperatures are modelled as too warm in CFSR and vice versa. As a result of this, CFSR comparisons have a lower correlation with observations than any reanalysis other than JRA55C.





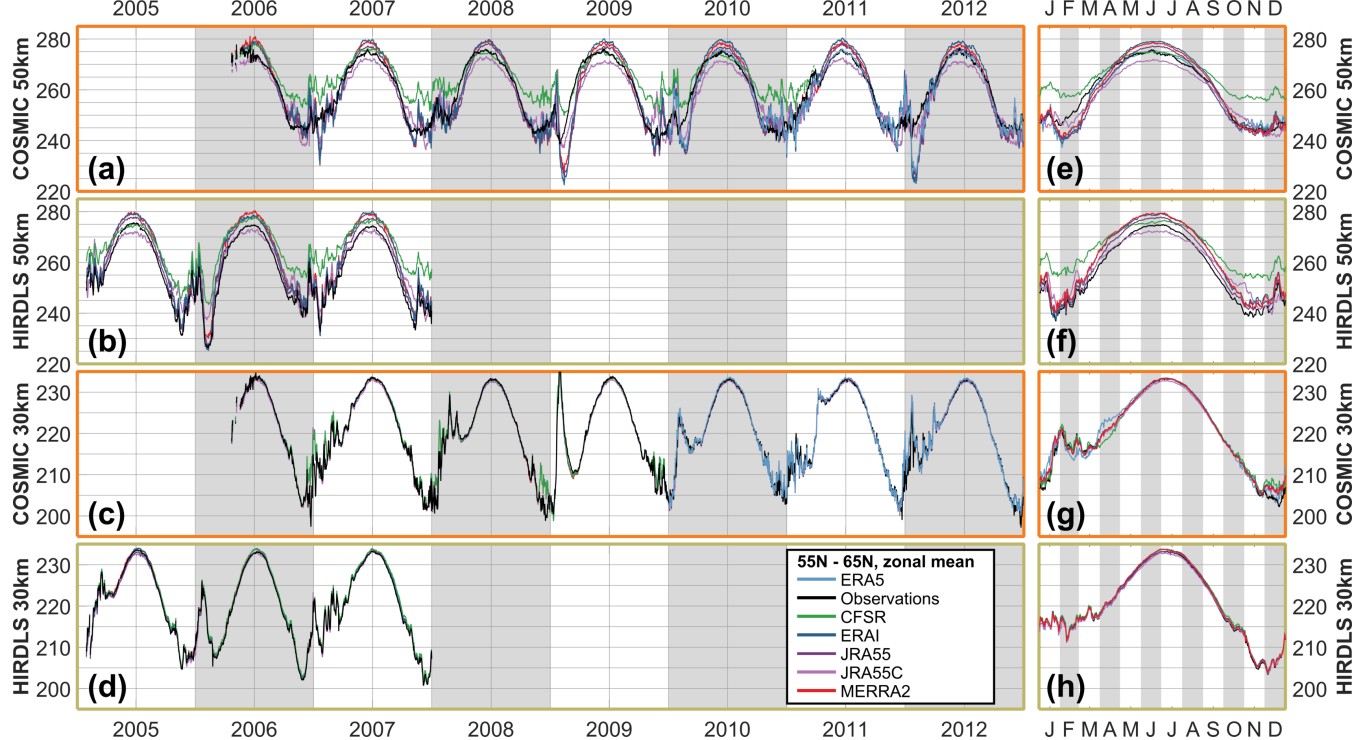

**Figure 9.** Time series of observed and sampled-reanalysis temperature, averaged over $55°N - 65°N$, at (a,b,e,f) 50 km (c,d,g,h) 30 km altitude. Panels (a-d) show time series over the period 2003-2012, and panels (e-h) the annual cycle derived from the extended time series.

## 7 Time Series Comparisons

To further characterise the differences between the observations and observation-sampled reanalyses, Figures 8, 9 and 10 show zonal mean time series of each dataset at the 30 km and 50 km altitude levels for the equator, 60°N and 60°S latitude bands

5  respectively. COSMIC, HIRDLS and SABER are all shown at the equator, but SABER is omitted from the other two latitude bands since it only provides coverage for part of the year. For each comparison, the main panel shows a time series over the full period of comparison, while the smaller panel at right shows an annualised comparison over the whole period.

In all cases, and consistent with Section 6, a much closer agreement is seen between the observations and synthetic reanalysis temperatures at the 30 km level than then 50 km level. Some biases are clearer in this format however. In particular, relative to

10  the reanalyses, SABER (30 km, equator) exhibits a high bias ∼1–2 K all year, while HIRDLS (50 km, equator) exhibits a low bias ∼1–3 K between July and January.

Key geophysical features are clearly resolved in all observations and reanalyses. A very clear annual cycle is seen in the 60°N and 60°S comparisons at both heights, a quasi-biennial pattern in the 30 km equatorial time series, and a mixed annual/semi-annual signal in the higher-altitude equatorial time series. While these features are extremely large and should be well-resolved





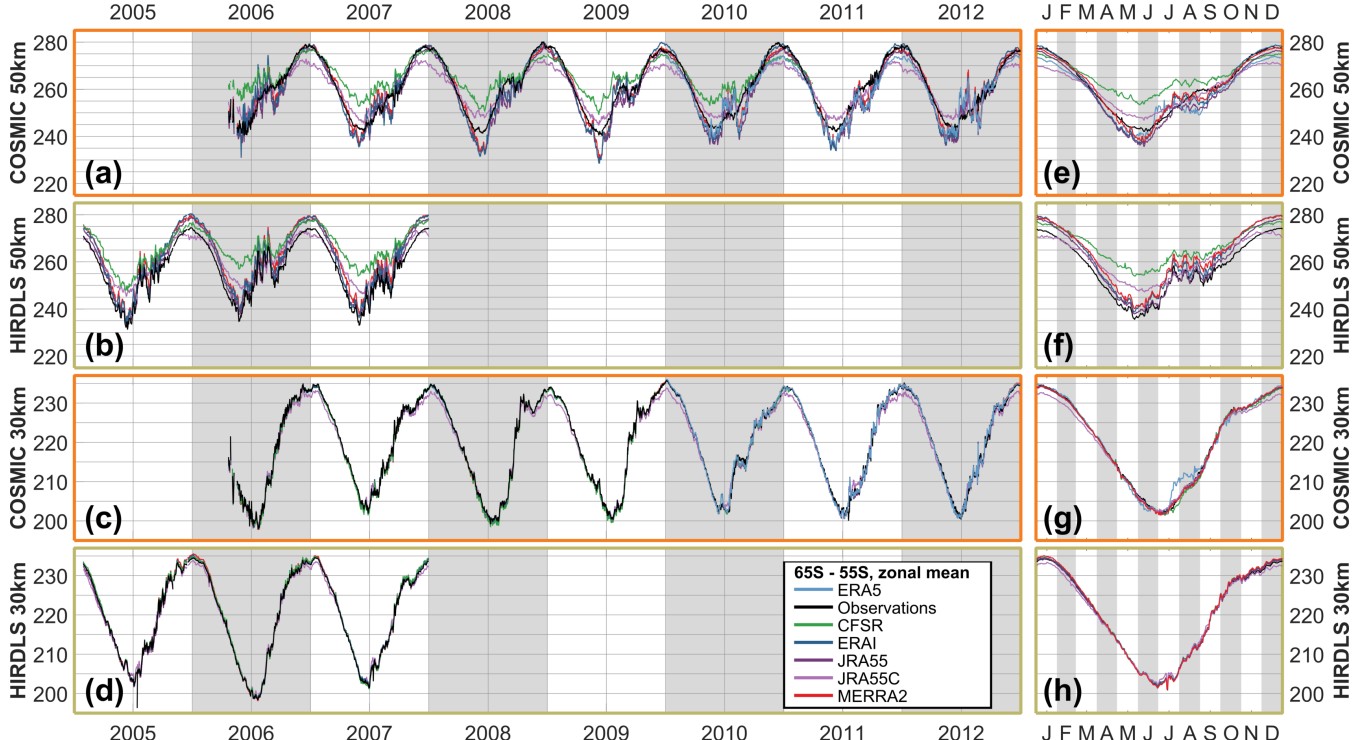

**Figure 10.** Time series of observed and sampled-reanalysis temperature, averaged over $55°S - 65°S$, at (a,b,e,f) 50 km (c,d,g,h) 30 km altitude. Panels (a-d) show time series over the period 2003-2012, and panels (e-h) the annual cycle derived from the extended time series.

in all datasets, it is encouraging to see how strong the temporal agreement is. A possible exception is CFSR at 50 km, which in both hemispheres has trouble resolving winter temperature minima at 50 km altitude.

Less consistency is seen in how well the different datasets reproduce the dramatic temperature variations associated with

5    sudden stratospheric warmings (SSWs). The SSWs of January 2006, January 2009, March 2010 and January 2012 are clearly visible at both altitudes, but the warmings of February 2007 and February 2008 (Butler et al., 2017), while visible upon close examination, are less clear. However, this is the case in both the original observations and the sampled reanalyses, and is thus likely to be due to geophysical differences between the SSW events rather than instrument or reanalysis performance.

For the well-resolved SSWs, all datasets show a near-identical response at the 30 km level. However, at 50 km much more

10   spread is seen between different datasets. Considering first the 2006 warming with HIRDLS sampling (Figure 9(b)), the temperature peak in late January is very well reproduced across the different datasets, but the period of reduced temperatures after this peak shows very large inter-dataset differences. While the observations, JRA-55 and ERA-I all reach a low of 226 K in this period, MERRA-2 estimates a temperature of 231 K, JRA-55C 238 K and CFSR 244 K (recall that HIRDLS has a known cold bias at these altitudes, and thus the true value is likely to be a few Kelvin higher than the observed). Thus, we see a range of 18 K for a zonal mean estimate made using a relatively densely-sampling instrument scan pattern (HIRDLS typically measured





~300-400 profiles in this latitude range per day). Smaller differences are seen in the February 2007 warming, with only CFSR diverging significantly from the consensus temperature estimate.

Considering now the SSWs resolved in the COSMIC-sampled record at 50 km (Figure 9(a)), we again see the initial temperature peak being measured similarly across all the datasets, but the temperature trough afterwards being resolved very differently in each dataset. For the 2009 and 2012 warmings, the spread in temperature is similar to 2006 in HIRDLS-sampled data, with the exception that the observational dataset is one of the larger-valued datasets, consistent with the challenges of measuring cold temperatures with COSMIC (Section 6). ERA-5, which is only currently available post-2010, joins ERA-I and JRA55 in simulating very cold temperatures in the 2012 event.

## 8 Taylor Diagrams

To investigate how well the synthetic measurements from each reanalysis reproduce the true measurements for each instrument, Figure 11 shows Taylor diagrams (Taylor, 2001) for each limb-sounder sampling pattern at three altitude levels: 30 km, 50 km and 70 km. For each panel, the standard deviation of the observational dataset is shown as a black circle on the horizontal axis, and acts as a 'true' estimate which the models are attempting to approximate. Each sampled reanalysis is then shown as a marker somewhere in the quadrant, with the angular distance between this marker and the black circle encoding the Pearson linear correlation between the two datasets, and the linear distance between the marker and the black circle (grey circles) showing the root-mean-square difference (RMSD) between them. The correlation and the standard deviation in each case are computed using the entire dataset, across all time and at all locations.

If the observations were perfectly true, these diagrams would allow us to quantitatively estimate reanalysis performance. The assumption of observational truth is however somewhat problematic given the technical challenge of stratospheric remote sensing measurements, and this distinction between measurement and truth must be borne in mind when considering the information these diagrams present: these diagrams are best thought of as describing how *similar* the reanalyses are to the observations rather than how *true* they are. Sections 10.1 and 10.2 below will generalise these results to all combinations of true and synthetic observations.

Considering first the 30 km level (Figures 11(f-h)), we see very close correspondence between all reanalyses and all three instruments. RMSDs in all cases are less than 3 K, and correlations greater than 0.95 (consistent with the scatterplots in Section 6). Interestingly, in all cases the reanalyses are all tightly clustered around a location close to each other but different to the observations, which will be discussed further below.

At the 50 km level (Figures 11(c-e)) differences start to emerge. Clear differences are seen in the standard deviations, with CFSR consistently the lowest and the observations consistently the highest. This is consistent with expectations: in Sections 6 and 7 we saw that CFSR temperatures were comparatively narrowly distributed at this level, while real observations are likely to have a small but consistent random noise component which will increase their standard deviation even in the case of otherwise perfect agreement. JRA-55C has the largest RMSD from the observations at this height in all sampling patterns, again consistent with Section 6.





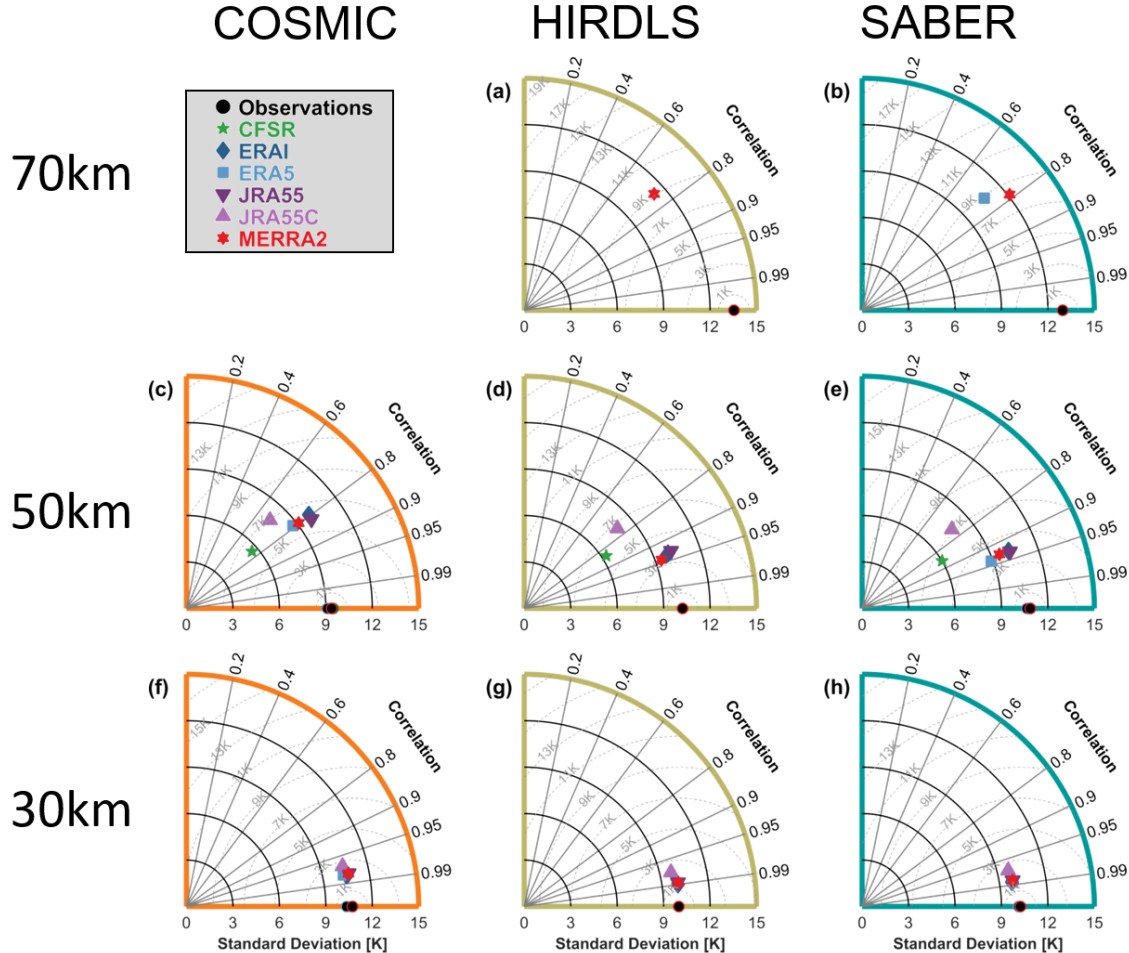

**Figure 11.** Taylor diagrams for each reanalysis-instrument pairing. For each panel, the radial axis shows the standard deviation of each sampled-reanalysis dataset (indicated by coloured symbols), and the curved axis indicates the correlation of that dataset with the equivalent observational dataset. Black circles indicate the observational datasets themselves, which by definition perfectly autocorrelate. Grey arcs show the root-mean-square-difference between the observational and sampled datasets. Multiple black circles are plotted on each panel, outlined with colours; this is because each reanalysis covers a slightly different time range, and thus there is a small amount of variability in the observational standard deviation; RMSD arcs are based upon the MERRA-equivalent observational standard deviation in each case (i.e. the red outlined circles). Each row shows a different height level, and each column a different instrument.

Finally, at the 70 km level (Figures 11(a-b)) we compare HIRDLS to MERRA-2 and SABER to MERRA-2 and ERA-5. The RMSD between observations and models is typically ∼9 K, with correlations ∼0.75. The ability of the reanalyses to reproduce the observational record is broadly comparable between HIRDLS and SABER.




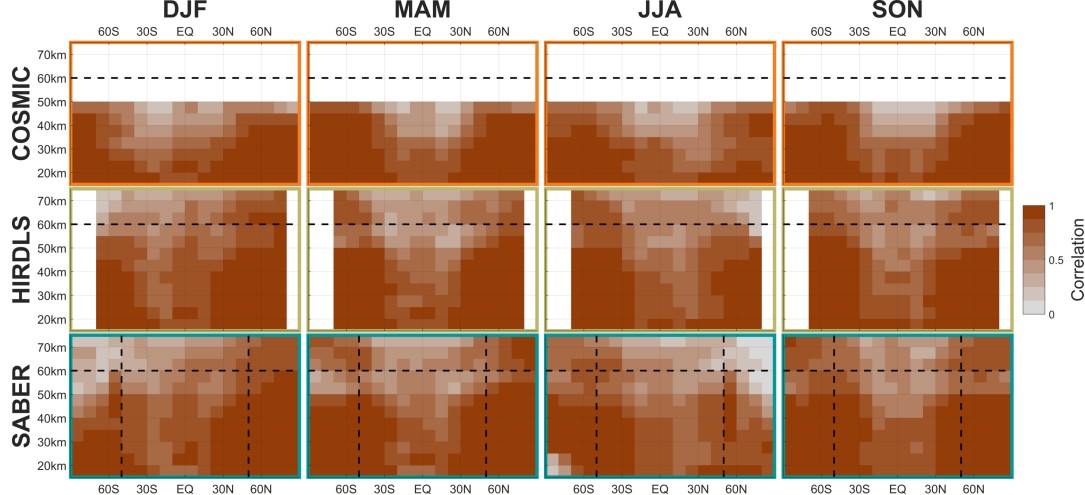

**Figure 12.** Correlations between Multi-Reanalysis Mean (MRM) sampled temperatures and each instrument, decomposed by latitude, altitude and season. Dashed line at 60km altitude indicates the approximate height of the transition from a six-reanalysis MRM to a two-reanalysis MRM; dashed lines at 50°S and 50°N indicate boundary full and partial SABER coverage. See supplementary figures S1-S6 for individual satellite-reanalysis comparisons.

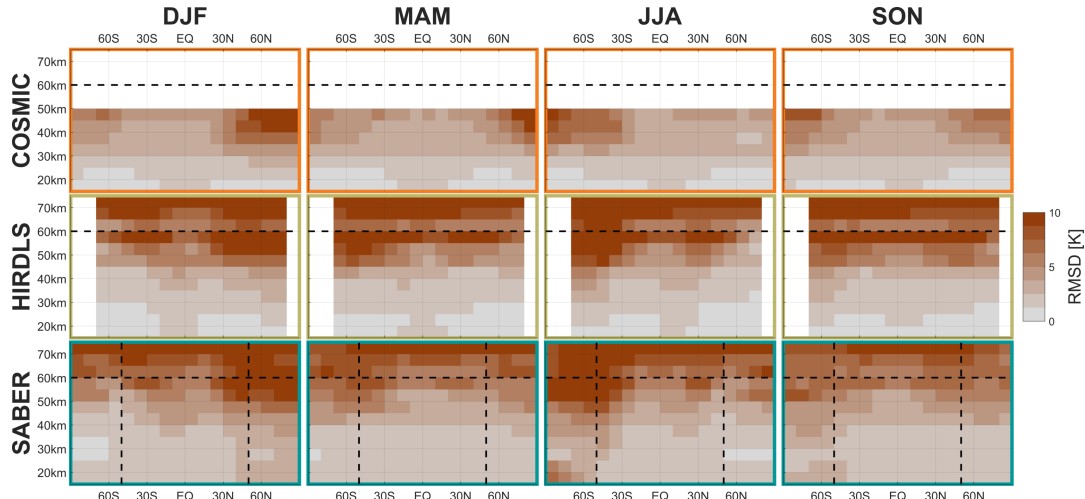

**Figure 13.** Root-mean-square differences between Multi-Reanalysis Mean (MRM) sampled temperatures and each instrument, decomposed by latitude, altitude and season. Dashed line at 60km altitude indicates the approximate height of the transition from a six-reanalysis MRM to a two-reanalysis MRM; dashed lines at 50°S and 50°N indicate boundary full and partial SABER coverage. See supplementary figures S7-S12 for individual satellite-reanalysis comparisons.



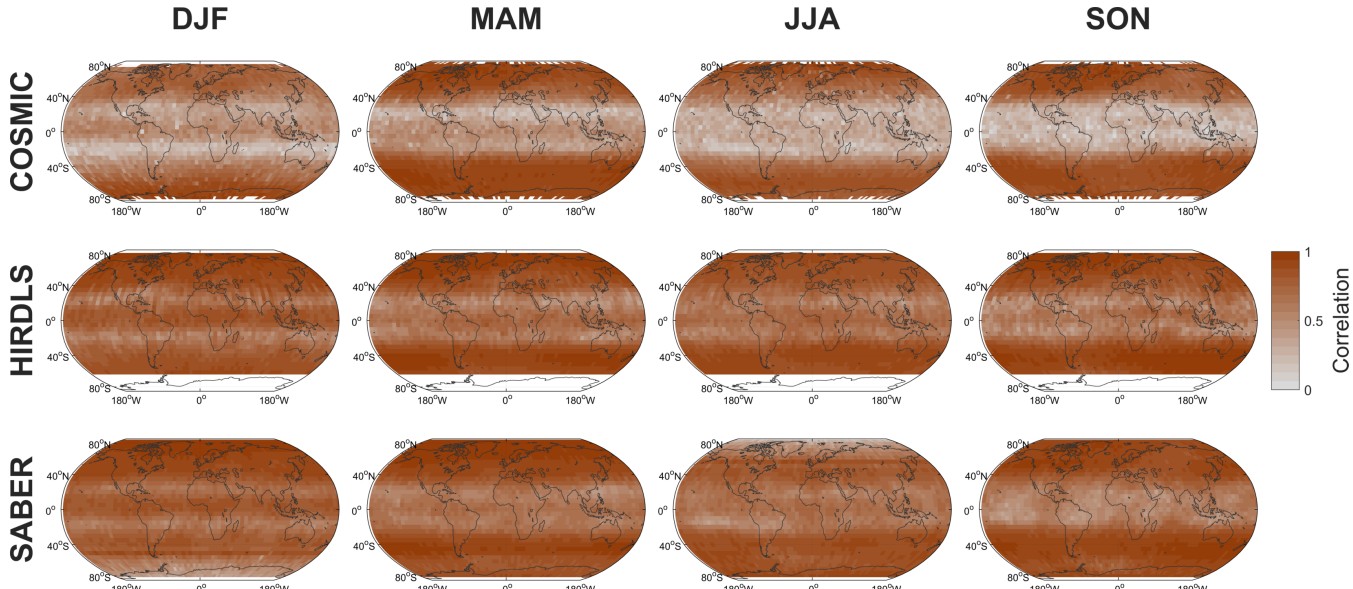

**Figure 14.** Maps at the 50 km altitude level of the correlation between MRM sampled temperatures and each instrument.

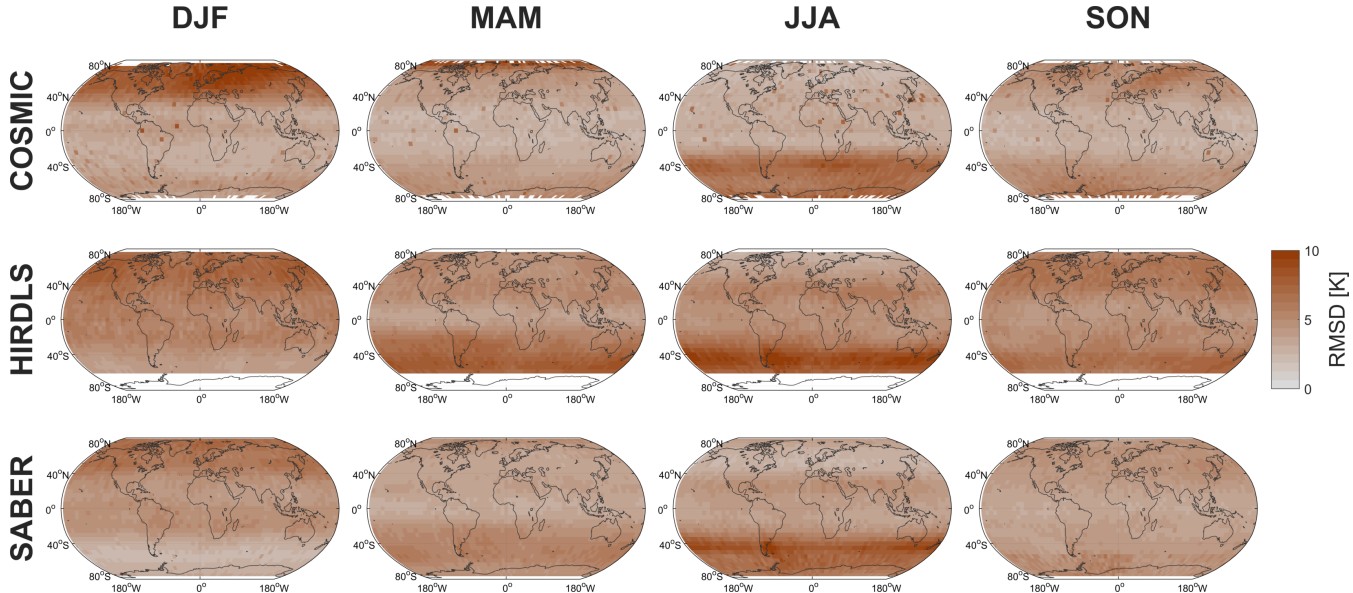

**Figure 15.** Maps at the 50 km altitude level of the RMSD between MRM sampled temperatures and each instrument.

## 9   Correlations by latitude and height

To further investigate the relative differences between our synthetic and true observations, Figures 12 – 15 decompose the data as functions of (Figures 12 – 13) zonal-mean latitude and altitude and (Figures 14 – 15) geographic location at 50 km, in all





cases decomposed by season. The 50 km level is deliberately selected here in preference to the 30 km level due to the wider spread of the above comparisons.

For reasons of space, these figures compare the observational data to a multi-reanalysis mean (MRM) rather than the individual reanalyses. This MRM is constructed as the arithmetic mean of all six synthetic measurements corresponding to each true measurement, i.e. it is at the individual measurement level. We also include full latitude-height comparisons to each individual reanalysis as Supplementary Figures S1 – S12; we note that each of these individual comparisons is very similar to the MRM comparison, and thus that the MRM figures characterise general differences between the observations and reanalyses well.

We consider first correlations and RMSDs as a function of latitude and altitude, Figures 12 – 13 and S1 – S12. Correlations are highest, and RMSDs lowest at high latitudes and low altitudes in all seasons. Lower correlations are seen in tropical and equatorial latitudes, but with the same vertical trend. Lower correlations are observed in all cases at the summer pole, falling to near-zero correlation for the southern summer pole in mesospheric SABER comparisons. This is presumably related to the well-known cold-pole problem (e.g. McLandress et al., 2012; Hindley et al., 2015; Wright et al., 2017; Garfinkel and Oman, 2018) which affects model temperatures over the winter pole due to issues resoving gravity-wave fluxes, compounded by a limited volume of assimilative constraints. Interestingly, these low correlations correspond to slightly lower RMSDs relative to other values at the same altitude level.

The maps, Figures 14 – 15 generally show similar results. However, one key feature is much clearer in this figures; namely that, while correlations are low across the tropics, the region directly above the equator (latitudes $<\sim 10°$) exhibits stronger positive correlations than the rest of the tropics in all three comparisons and in all four seasons. Since model development often focuses on reproducing the stratospheric QBO at these latitudes as a proxy for accurate atmospheric wave simulation, this may indicate that this region is especially well-tuned by comparison to the surrounding latitudes.

## 10   Cluster Analysis

### 10.1   Cluster Circles

An interesting feature of the Taylor diagrams presented in Section 8 is that, at each height level and for each instrument sampling pattern, a clear cluster is seen consisting of all the reanalyses (except CFSR and JRA55C at 50 km), as opposed to a wide scatter. The MRM comparisons presented in Figures 12 – 15 and the individual-reanalysis comparisons presented in Figures S1-S12 also show a close geographic correspondence in the magnitude and pattern of the difference between true and synthetic measurements.

However, this does not itself tell us if the reanalyses are similar to each other. It is plausible that they could all differ from the observations to a similar quantitative degree but differ strongly from each other at the same time. This is an important distinction, and requires further clarification. Accordingly, we have also computed the correlation and RMSD between each pair of datasets for each sampling pattern, which we present in Figures 16 and 17.

In both figures, each panel shows all pairwise comparisons for a given instrument sampling pattern at one of three height levels. Individual datasets (both reanalysis and observational) are evenly distributed around the exterior of the shaded circle.





Figure 16. Pairwise correlation between each dataset at three height levels.

Lines then join each pair of datasets, the colour and width of which indicate the correlation coefficient (Figure 16) or RMSD (Figure 17) between that pair of datasets using that sampling pattern. For example, Figure 16(f) shows the correlations between all synthetic and real measurements using the HIRDLS sampling pattern at 30 km altitude, while Figure 17(b) shows the RMSD between each pair of datasets using the COSMIC sampling pattern at 50 km altitude. Observational datasets are shown at the right of each panel.

Thick blue lines show the best agreement between a pair of datasets, and thin red lines the worst. In each case, these lines are scaled across the range of values measured for that panel: specifically, thin red lines show the lowest correlation (or highest RMSD) measured, thick blue lines show the highest correlation (or lowest RMSD) measured, and the other lines



**Figure 17.** Pairwise RMSD between each dataset at three height levels.

are scaled linearly into the range between these values based on their numerical value, as opposed to their ordering. This internal normalisation is chosen due to the large inter-panel differences in value (as seen in Sections 6, 8 and 9). Full numerical correlations and RMSDs are provided for each pairing as Supplementary Tables S1-S14.

5     Firstly, we note that the worst inter-dataset pairings are almost always with JRA-55C, in each case shown at the bottom of the circle. CFSR also compares poorly at the 50 km level. These tendencies are consistent with our previous results and expectations.

    The second feature we note is that no observation-reanalysis relationship is ever the strongest relationship within that panel. This is the case for all sampling patterns and for both correlation and RMSD analyses. For RMSD comparisons, this is clear



from Figure 17, since no observational point has a thick blue line connecting it to any reanalysis. For the correlation compar-isons, Figure 16, this is less clear since two connections (HIRDLS to MERRA-2 at 50 km and ERA-5 to SABER at 50 km) are shown as thick blue lines; however, even in these two cases the measured value (Tables S10 and S11) is the lowest correlation

of the thick-blue-line pairings.

This is especially interesting in the case of COSMIC at 30 km altitude, panel (e) of both figures. While HIRDLS and SABER are independent of the reanalyses and COSMIC is itself unreliable at 50 km altitude due to noise effects, at 30 km altitude COSMIC is both highly reliable at a technical level, and, crucially, is actually assimilated by every model considered. Given this, we would expect all the synthetic measurements to be highly similar to COSMIC retrieved temperatures, with which they

share a sizable fraction of the detection and analysis chain. Even more unexpectedly, COSMIC is actually the worst-correlated dataset of the seven considered and has the highest RMSD, even including JRA-55C. We discuss this further in Section 10.2.

Less surprisingly, but still interestingly, the same is largely true for SABER and HIRDLS comparisons. Compared to the relationships between most reanalyses, the RMSD between observational and reanalysis datasets is much larger and the corre-lations lower. This implies a significant dissimilarity in every case between real and synthetic observations. While some of this

difference could be due to limited instrumental precision, this does not explain all or even the majority of the observed effect size: typical RMSDs between synthetic and real observations (Tables S1-S14) are ∼1-2 K at the 30 km level rising to ∼4-9 K at the 50 km and 70 km levels, while instrument precision is typically ∼0.5 K.

## 10.2 Cluster Analysis

Section 10 suggests that there are meaningful differences between observational and synthetic datasets, in terms of both form

(i.e. correlation) and magnitude (i.e. RMSD). However, the methods we have used so far cannot distinguish between two possible reasons why this may be the case:

1. the reanalyses are overly similar to each other relative to the differences between observations, or

2. the observations are overly similar to each other relative to the differences between reanalyses

This is an important distinction, with possible implications for the tuning of both reanalysis model and satellite retrieval

development. In the former case, the models may be too tightly tuned against each other, while in the latter case the different instrumental retrievals may be too tightly tuned against each other. To assess this, we must compare satellites instruments against each other as well as against their synthetic equivalents. Accordingly, we have produced a subset of the full dataset consisting only of measurements co-located between each pair of instruments, allowing us to make direct comparisons across the full range of instrument-reanalysis combinations.

We define profiles as co-located if they are within 100 km and 15 minutes of each other, following the argument of Wright et al. (2011)). For this subset of the data, we then compute the RMSDs and correlation coefficients between the samples from all twenty datasets, i.e. the three observed datasets and the seventeen sampled datasets, and use these to compute a hierarchical tree. As a caveat, we note clearly that the set of measurements is not exactly identical in each combination, since measurements from all three satellite instruments only very rarely coincide; thus, the samples correlated between (e.g.) HIRDLS-as-CFSR



and SABER-as-JRA55 will not be the same set of samples as (e.g.) HIRDLs-as-CFSR and COSMIC-as-JRA55. However, there a sufficient number of pairwise correlations between each observational track (of order thousands), and the samples are geographically sufficiently widely distributed, that all of the correlations and RMSDs can be collected into a single hierarchical

tree with traceability across the full set. This co-location was carried out at the 30 km altitude level, and our subsequent analysis and discussion focus only on this level for brevity. The same analysis was performed at 50 km altitude and gave broadly similar results, except that differences with observed COSMIC data and with CFSR synthetic temperatures were much larger, consistent with the issues described previously.

Once the data have been subsetted, we carry out a hierarchical cluster analysis (Hastie et al., 2009; Wright et al., 2013) of

these correlation and differences. We first combine RMSDs and correlations into a single composite metric of dissimilarity. Specifically, we define the maximum range of measured RMSDs and the maximum range of measured correlation coefficients to both equal one, and then combine them as

$$\text{Combined difference} = \text{Scaled RMSD} + (1 - \text{Scaled Correlation}). \tag{1}$$

A dataset pair with the minimum correlation and the maximum RMSD across the full set of combinations will thus score close

to two. This choice of a 1:1 weighting of RMSD and correlation coefficient is arbitrary, but our results show minimal sensitivity to varying their fractional weight, and the same conclusions are reached using either metric individually.

We then perform a hierarchical clustering analysis on the combined difference between each possible pair of sampled and observed datasets, using the single (i.e. minimum difference) linkage and a Euclidian distance metric. The ordering of our results is invariant to the choice of linkage and distance metric used over a wide range of feasible options including the

complete, average and centroid linkages, and the Ward and standardised-Euclidian linkages. This analysis essentially partitions the inter-dataset comparisons into a hierarchical tree, starting at the lowest possible difference between any two datasets and then producing a new branch when a rising difference floor reaches a critical level of dissimilarity. The results of this are illustrated as a dendrogram, shown in Figure 18.

Although there are some subtleties to a precise interpretation, to first order this diagram can be conceptualised as showing

the relative difference between each datasets, with the maximum height reached on the path through the tree which joins them indicating their relative difference from each other (in bioscience applications, this separation would be their cophenetic distance). Horizontal distance does not imply any information, and the ordering is chosen purely to produce a simple tree. We thus obtain an idea of the relative overall difference for all possible combinations of sampled and real datasets.

We first note that each reanalysis forms a distinct branch of the tree, with all sampled forms of any given reanalysis tightly

connected. This suggests that the smallest differences are between different forms of the same dataset. This is an intuitively sensible result, since we would expect any given model to correlate well with itself.

Beyond this, we see three main groups of results. The first of these is JRA-55C, at the right of the panel, where all three forms of the dataset are closely related to each other (i.e. they join the hierarchical tree only a short way up the panel), but are relatively much more dissimilar from any other dataset (i.e. they join the rest of the tree at the extreme top of the panel). This tells us that, again in relative terms, the sampled JRA-55C data has significant dissimilarities with all other reanalyses and




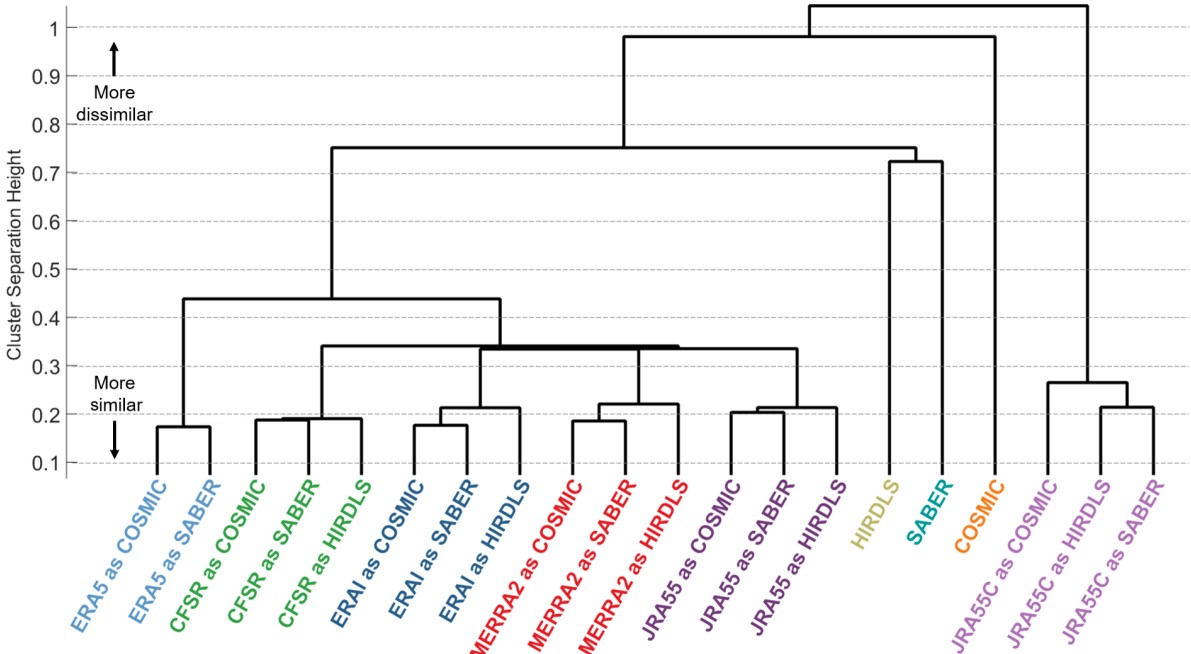

**Figure 18.** Dendrogram illustrating results of cluster analysis on co-located measurement-pairs. See text for details.

with all the instrumental datasets. This is again an intuitively sensible conclusion, due to the lack of observational constraints
at these altitudes in JRA-55C.

The second main group of results is the set of observational datasets, i.e. COSMIC, HIRDLS and SABER, to the left of the
JRA-55C cluster. The defining characteristic of this group is not their relative closeness but instead the exact opposite. Each of
the observational datasets is highly dissimilar from both the others and from all the reanalyses, regardless of sampling pattern.
Of the three, HIRDLS and SABER join each other in the tree most closely, while COSMIC is more dissimilar to the other
observational datasets but less so than JRA-55C.

Our final, third, group is of the full-input reanalyses. This is the majority of the tree, extending from the far left rightwards.
All of these datasets join the tree at a point approximately halfway up the panel relative to where HIRDLS and SABER
join it, suggesting that this group is significantly more similar to each other (on the basis of our distance metric) than to
any observational dataset. Within this group, ERA-5 samples form a distinct subset, while the others are all relatively tightly
clustered to each other.

From this analysis, we conclude that, of the two options outlined at the start of this section, option (1) is the more likely, i.e.
the set of (full-input) reanalyses exhibits a high degree of internal similarity relative to any individual observational dataset.
Indeed, our analysis suggests that each of the observational datasets is more different to from any other observational dataset
than any full-input reanalysis is from any other full-input reanalysis.



Of special note is the relatively large dissimilarity between COSMIC and any other dataset. Since all of the full-input reanalyses assimilate low-level COSMIC data, this suggests either that the COSMIC temperature retrieval introduces large additional errors relative to the assimilated form, or that the relative importance of COSMIC data in the reanalysis schemes used is too low.

## 11  Summary and Conclusions

In this study, we develop and apply a method of sampling output reanalysis temperature fields to produce synthetic satellite observations. This allows us to carry out a like-for-like comparison between final-output reanalysis products and final-output satellite data products, i.e. at the end of their respective analysis chains, where a typical user would encounter the data.

We first compare the synthetic measurements produced using this approach to a simplified scheme where the reanalysis is simply interpolated to the centre of the satellite measurement volume. We show that, with the current generation of reanalyses, the more complex approach is always required when comparing to AIRS, almost never required when comparing to HIRDLS, and required in equatorial regions and regions of high gravity wave activity but not otherwise for COSMIC and SABER.

We then assess the relative differences between each of the three limb sounder datasets (i.e. COSMIC, HIRDLS and SABER) and six modern reanalyses at the 30 km and 50 km altitude levels. Unsurprisingly, agreement between observations and reanalyses is significantly better at the 30 km level than the 50 km level: all-latitude all-time correlations between observations and full-input reanalyses (i.e. excluding JRA-55C) range from 0.979 – 0.990 at the 30 km level, reducing to 0.842 – 0.942 at the 50 km level (also excluding COSMIC at 50 km, which is known to have measurement difficulties at this altitude).

The reanalysis and observed datasets generally show excellent agreement at the zonal mean level. Exceptions typically occur at higher altitudes, the largest of which is a general mismatch following sudden stratospheric warmings.

Inter-dataset correlations are lowest (i) in the tropics at all heights except for the region immediately above the equator, (ii) at all latitudes at high altitudes, and (iii) at high altitudes over the summer pole. RMS differences are highest at high altitudes and over the winter pole.

Finally, we use both intra-sampling-pattern all-dataset and inter-sampling-pattern colocated-profile-only correlations and RMSDs to investigate how similar the reanalyses and individual observational datasets are to each other at a bulk level. Our results show strong evidence that, with the exception of JRA-55C, the variability between any pair of reanalysis datasets is significantly less than that between any two observational datasets or between any observational dataset and any reanalysis, even in the case of COSMIC where the reanalyses assimilate that data. This may be evidence of over-tuning of reanalysis models against their comparators, and if so presents significant implications for future development of these models.

*Code and data availability.* The original reanalysis and satellite datasets are available from their respective sources, as described in the acknowledgements. All code developed for this study is available as Matlab scripts on request from the lead author, and the intermediate data products used can be generated using these scripts.





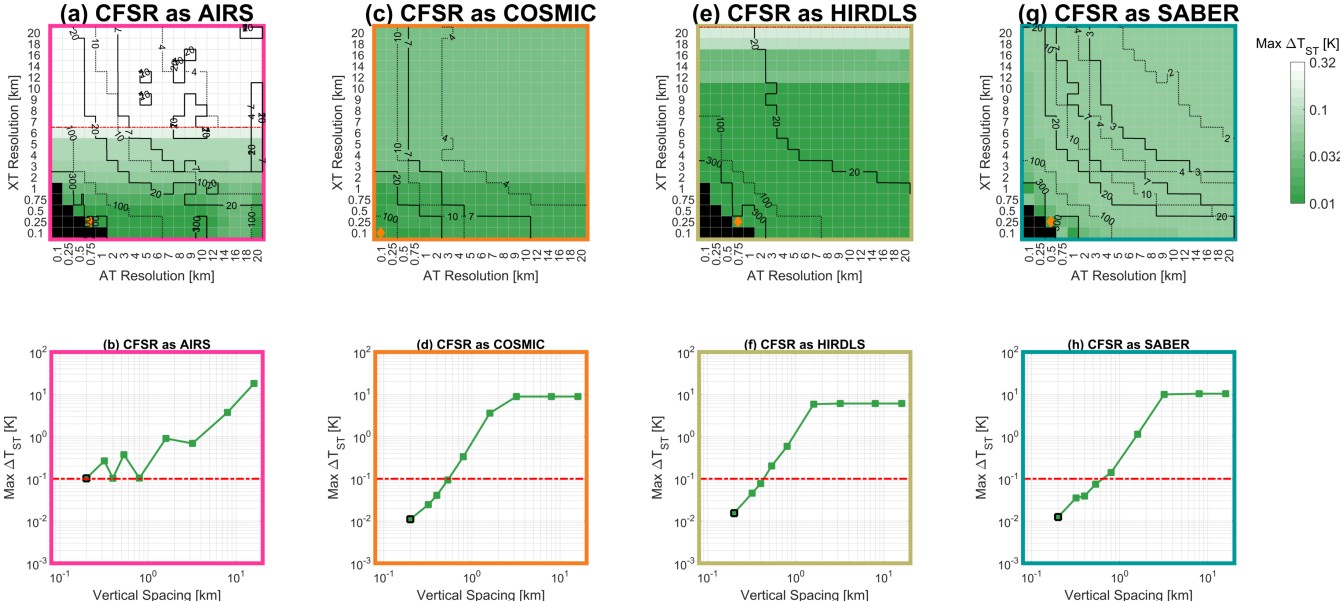

**Figure A1.** Sensitivity testing results. The top row shoes the results of horizontal sensitivity tests, plotted as the maximum difference in temperature between the test and a baseline indicated by the orange diamond; line contours show required runtime. The lower panel shows the results of vertical sensitivity tests, plotted as the maximum difference from a reference level of 1/160 decades vertical spacing.

## Appendix A: Selection of Fine Grid Parameters

Our results are potentially highly sensitive to the fine grid spacing used in the analysis. Too coarse a spacing will lead to results which do not represent the reanalysis temperature in the sampling volume accurately, while too fine a grid will be prohibitively computationally expensive. To assess the best fine grid spacing to use, we performed a series of sensitivity tests prior to carrying out the main analysis, using data from the 1st of January 2007 as a proxy for the mission as whole. Figure A1 shows the results of these tests for each instrument sampled from the CFSR reanalysis. The other reanalyses were also sensitivity-tested (using data from the first of January 2012 for ERA-5), but produced very similar results and are consequently omitted for brevity.

For each instrument, the top panel shows the results of our horizontal sensitivity test. In each case, the vertical axis shows the spacing of the fine sampling grid in the minor ($Y$) direction, and the horizontal axis the grid spacing in the major ($X$) direction. For each test in this panel, a fixed vertical fine grid spacing of 1/80 decades of pressure ($\sim$200 m) was used. We imposed a maximum runtime of 6 hours per day of data on this test, and black squares indicate tests which did not sample a complete day of data within this time.

We first define a baseline result, chosen as the highest-resolution test to run within 6 hours. This baseline is assumed to best represent an optimally-sampled reanalysis temperature field, and is indicated by the orange diamond on each panel. For each other test, we then find the sensitivity-testing temperature difference $|\Delta T_{ST}|$ between every sample extracted that day in the test and in the baseline, and plot the maximum value of this difference. By definition, the baseline test thus has a $\Delta T_{ST}$ of 0 K,




with all other tests having positive values. The values shown thus represent the largest difference across the whole globe at all heights between the highest-resolution fine grid used and the fine grid being assessed in that test. Line contours indicate the

runtime required to achieve each result, in minutes.

Intuitively, it seems odd at first that AIRS-L1 sampling is much more sensitive to the nominally-minor $Y$ direction than the nominally-major $X$ direction, since the input horizontal weighting functions are circularly symmetric. However, when the functions are rotated vertically to account for the across-track scanning pattern of the instrument, a significant asymmetry arises in the $Y$ direction, causing this increased sensitivity. Otherwise, the dependencies are as one would intuitively expect,

with an extremely strong $Y$ sensitivity for the very narrow COSMIC weights, weaker $Y$ sensitivity for the broader HIRDLS and SABER, and the weakest dependency in all cases on the $X$ direction for all instruments.

The lower panels show equivalent results for vertical sensitivity, in each case performed at the chosen horizontal sensitivity. These are plotted as differences from the finest vertical spacing used (1/160 decades of pressure, $\sim$100 m). The results are shown on logarithmic axes, and thus the 1/160 decade spacing, which by definition has a $\Delta T_{ST}$ of 0 K, is not shown. The 1/80

decade spacing used in the horizontal tests is indicated by the black outlined point.

Finally, the same analysis was repeated using data pre-smoothed by 5 km in the vertical, to counter any effect of random noise. No significant change was observed.

Based on these results, we choose our final fine sampling grid for each instrument as an empirical balance between time and small $\Delta T_{ST}$. These are the values shown in Table 2, and have a maximum $\Delta T_{ST}$ in each case of less than 0.1 K, i.e. significantly

below satellite sensitivity.

## Appendix B: Full Sampling vs Single-Point Sampling

In addition to our full-3D approach, our analysis routine also samples the reanalyses by simple linear interpolation to the centre of the measurement volume. This allows us to make a direct comparison to our more expensive approach, to asses how useful it is. We refer to these approaches as, respectively, the 'SPA' and 'full' approaches and to the difference between the resulting

temperature estimates as $\Delta T_{SPA}$. We note clearly that the magnitude of $\Delta T_{SPA}$ is not a metric of reanalysis quality (although it does inherently arise out of reanalysis resolution and small-scale variability, both of which may relate to quality), but is instead intended to determine whether the full sampling approach is justified for comparing satellite data to said reanalysis.

### B1 Overall Differences

Figure B1 shows the results of this analysis at a bulk all-measurements level, plotted as cumulative distributions of $\Delta T_{SPA}$ for

all synthetic measurements at all height levels.

For all reanalyses sampled as COSMIC, HIRDLS and SABER, mean differences between the SPA and full approaches are $\leq$0.17 K in all cases. This is better than the measurement error of these instruments, the precisions of which are estimated as $\sim$0.5 K for HIRDLS and COSMIC and $\sim$0.8 K for SABER. For AIRS-L1, differences are larger, with mean $\Delta T_{SPA} \sim$6–8.5 K compared to a precision $\sim$0.7 K, i.e. an order of magnitude larger than instrument noise.





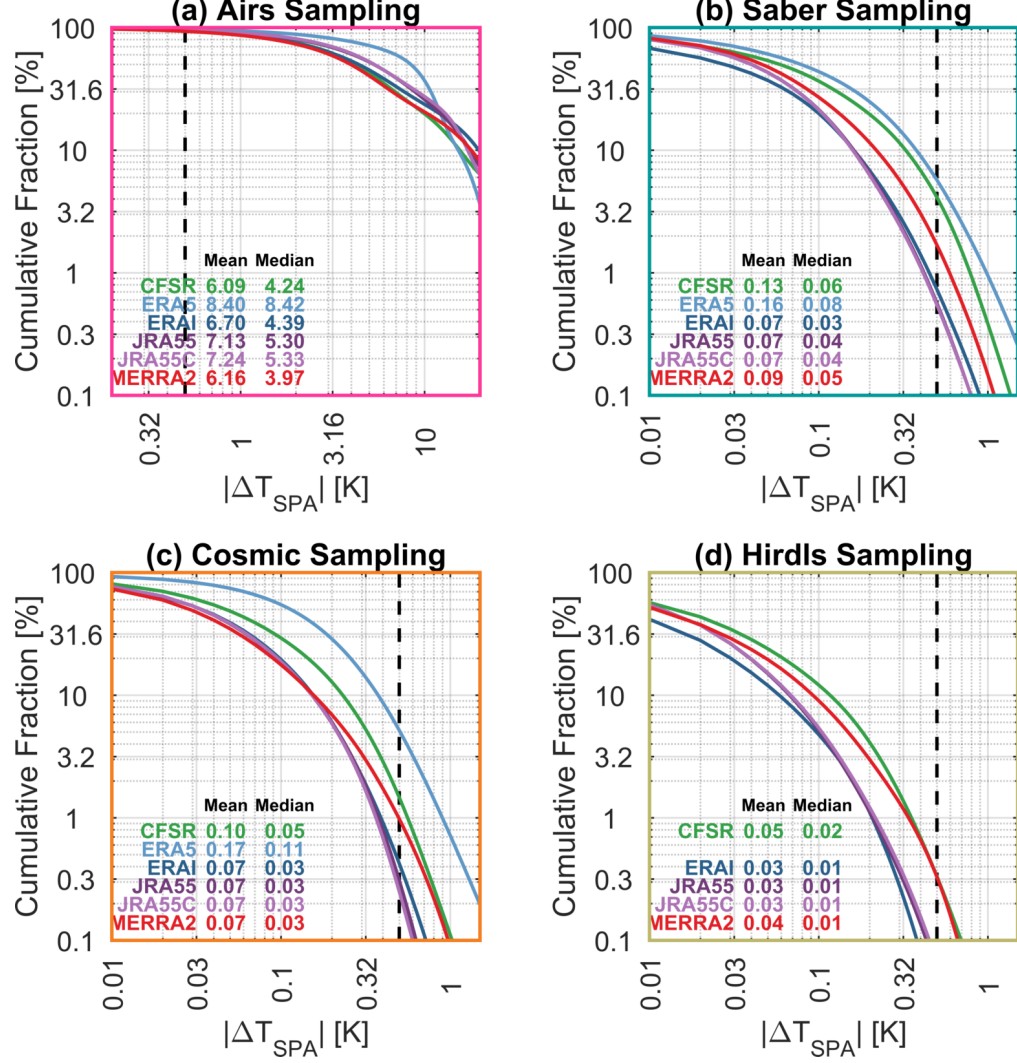

**Figure B1.** Cumulative distributions of the absolute difference between full sampling and single-point sampling. Note that panel (a) has a different abscissa to the other three panels, due to the very different width of the distribution.

These results are consistent with the broad nature of the sensing methods used. Atmospheric temperature typically varies much more rapidly in the vertical than the horizontal, and thus the SPA will tend to approximate the fine vertical sensitivity of
5    a limb sounder better than a nadir sounder's coarser vertical sensitivity. We note also that our AIRS sampling is only at a single height level, whereas the limb sounders are sampled across a range of heights, and that this may affect the results, particularly since the AIRS weighting functions we use are centred close to the ozone-induced temperature peak usually present in the upper stratosphere.



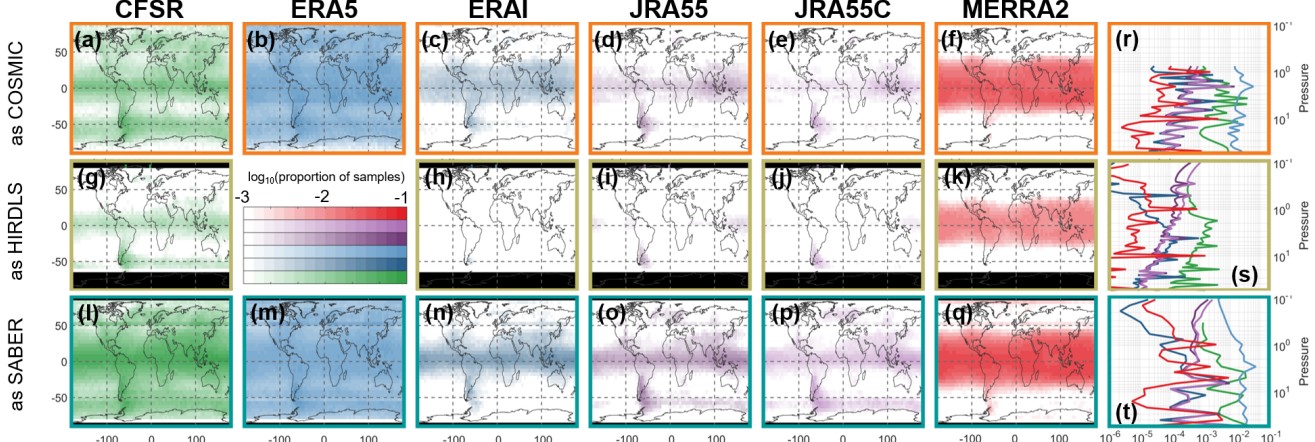

**Figure B2.** Geographic variability of $\Delta T_{SPA}$. The leftmost six columns show maps of the column-integrated proportion of samples with $\Delta T_{SPA} > 0.5$ K on a 2.5-degree grid, while the rightmost column shows the globally-integrated with $\Delta T_{SPA} > 0.5$ K at each height. Colour bars are the base-10 logarithm of the proportion of samples. Black regions on maps indicate a lack of coverage.

For the limb sounders, ERA-5 has the largest $\Delta T_{SPA}$, then CFSR followed by MERRA-2, and then a cluster consisting of ERA-I, JRA55 and JRA55C with broadly equivalent results. For ERA-5 and CFSR, this is presumably due to their very fine horizontal resolution, which means that information from more individual reanalysis points contribute in the full-sampling analysis. MERRA-2 then follows, with a final cluster made up of JRA55, JRA55C and ERA-I. The equivalence of ERA-I to these models in this analysis is interesting, as it has a much coarser horizontal resolution than the other models.

AIRS has a different ordering: ERA5 still exhibits the largest $\Delta T_{SPA}$, with the others five reanalyses more tightly clustered after it (note that the form of the ERA5 distribution changes at ∼8 K, which we believe to be due to the relatively small number of ERA5 points sampled). The difference in ordering compared to the limb sounder comparisons is likely due to the relatively low vertical resolution of some of the reanalyses at ∼4 km altitude. In particular, while CFSR has tightly-spaced levels in the lower stratosphere, vertical grid spacing is extremely coarse in the upper stratosphere (Figure 3), explaining why it goes from one of the smallest differences in the limb sounder case to more typical behaviour relative to other reanalyses in the AIRS case.

## B2  Geographic Variability of $\Delta T_{SPA}$

Figure B2 shows the geographic distribution of synthetic measurements with $\Delta T_{SPA} > 0.5$ K for the three limb-sounding instruments. We omit AIRS-L1 since the mismatches are present in almost every sample, and thus a geographic distribution does not provide useful additional information. 0.5 K is chosen as representing an approximate level at which $\Delta T_{SPA}$ is comparable to the uncertainty on the comparator satellite measurements, but the form of the results is broadly equivalent over a wide range of values.





Figures B2(a-q) show the column-integrated proportion of samples with $\Delta T_{SPA} > 0.5$ K for each reanalysis sampled as each instrument, computed on a $2.5° \times 2.5°$ grid. Most panels show a peak around the equator, particularly for MERRA-2 where

almost no large values are observed poleward of $30°$ in either hemisphere. This is consistent with the complex mesoscale dynamics of this region, where model-resolved Kelvin waves and mixed Rossby-gravity waves are a significant factor, as are gravity waves. The relatively fine height-layering of the quasi-biennial oscillation in this region may may also contribute.

We also see secondary maxima around the Southern Andes/Antarctic Peninsula region and across the Southern Ocean. These regions are known to have significant gravity wave activity at all length scales (e.g. Eckermann and Preusse, 1999; Jiang, 2002;

Alexander and Teitelbaum, 2011; Hindley et al., 2015), and the spatial correspondence with this activity is very precise, suggesting this as a possible mechanism for the mismatch between SPA and full sampling in this region. Maxima are also observed over other known gravity wave hotspots (e.g. SE Asia, Scandinavia, the convective equatorial regions), supporting this hypothesis. This does not necessarily imply that the reanalyses can resolve gravity waves at lengthscales shorter than the satellites observe. A more likely explanation is that the major axes of the satellite sampling volumes lie along the rising or

falling temperature gradient of a single wavefront at a moderate angle, which would manifest as a large $\Delta T_{SPA}$ value even if the wave was physically much larger in scale than the sampling volume.

Figures B2(r-t) show the globally-integrated proportion of synthetic measurements with $\Delta T_{SPA} > 0.5$ K for each reanalysis, as a function of altitude. There are slight height trends in some cases, but the differences are fundamentally dominated by noise and spikes rather than a strong height dependence or relationship to geophysical features such as the stratopause. Thus, we

conclude that $\Delta T_{SPA}$ exhibits no systematic height dependence at the global scale.

*Author contributions.* CJW designed the study, wrote the software, carried out the analyses, and produced the text and figures. NPH provided technical expertise and contributed to the scientific interpretation of the results at all stages of the process.

*Competing interests.* The authors have no competing interests.

*Acknowledgements.* CJW is funded by a Royal Society University Research Fellowship (ref. UF160545) and by Natural Environment Re-

search Council grant NE/R001391/1. NPH is also funded by this NERC grant. This research made use of the Balena High Performance Computing Service at the University of Bath. Reanalysis data were provided by the European Centre for Medium-Range Weather Forecasts (ERA-Interim and ERA-5), the NASA Goddard Global Modeling and Assimilation Office (MERRA-2), the Japan Meteorological Agency (JRA55 and JRA55-C), and the US National Centers for Environmental Prediction (CFSR). Satellite data were provided by NASA JPL (AIRS) NASA, Colorado University and NCAR (HIRDLS), NCAR (COSMIC), and GATS, Inc (SABER). CJW also acknowledges useful conversations with Dr James Anstey (CCCma) which helped to originally motivate this work, and with various colleagues at the 2017 SPARC Dynamics Workshop in Kyoto, Japan and at the EGU General Assembly 2018 in Vienna, Austria which helped to refine it.





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
