# Peer review of "How well do stratospheric reanalyses reproduce high-resolution satellite temperature measurements?"

_Atmospheric Chemistry and Physics, 2018_

## Referee Comment (RC1) · M. A. Geller (Referee) · 12 Jul 2018

This is a very important paper.  A common methodology in stratospheric research involves treating the results of reanalyses products as being analogous to atmospheric data from measurements.  There are several reasons for this.  One being the relative paucity of stratospheric measuerments, and another being the ease of use of reanalyses (i. e., no missing data, evenly spaced information ).  One need to examine the conclusions of papers such as Randel et al. (2004) and later works to see both the advantages and shortcomings of reanalyses representation of measurement data.  This paper by Wright and Hindley does a very careful mapping of reanalyses temperatures from several modern reanalyses (CFSR, ERA-5, ERA-Interim, JRA-55, JRA-55C, and MERRA-2) onto the HIRDLS, SABER, and AIRS satellite instrument and the COSMIC weighted measurement volumes for comparison with HIRDLS, SABER, AND COSMIC retrieved temperatures, as well as AIRS radiances at various altitudes in the stratosphere and mesosphere.  Comparisons are done using global data by means of scatter plots and the resulting correlations, and also by comparing time series for equatorial and high latitude zonal means at different altitudes.  In general, the principal conclusions are that the reanalyses closely reproduce the measurements at 30 km (correlations in excess of 0.98) except for JRA-55C with correlations about 0.97, despite the fact that JRA-55C assimilates no stratospheric data.  The correlations fall off substantially at 50 km, with the lowest correlations being for JRA-55C, followed by CFSR.  Interestingly,  the comparisons with COSMIC give lower correlations even though COSMIC stratospheric measurements are included in the assimilation process.

Examining time series comparisons shows several things.  SABER displays a high bias of 1-2 $^0$K at 30 km in equatorial latitudes, and HIRDLS shows a similar low bias at 50 km near the Equator.  All of the reanalyses show less fidelity to the observations at 50 km than at 30 km, but this is particularly true for the CFSR, with lesser fidelity to the observations during disturbed periods.

An interesting result shown in the Taylor diagrams is that all of the reanalyses are more similar to each other than to the observations.  A cluster analysis is carried out showing that the reanalyses are more correlated with each other than with any of the observations (including COSMIC, whose data is assimilated in all the reanalyses).  The suggestion here is that the reanalyses may be being "tuned" to each other excessively.  The HIRDLS, COSMIC, and SABER temperatures are noticeably less similar to each other than are the reanalyses temperatures to the other reanalyses.  The COSMIC temperatures are particularly dissimilar to the HRDLS and SABER temperatures.  The authors indicate that this "suggests either that the COSMIC temperature retrieval introduces large additional errors relative to the assimilated form, or that the relative importance of COSMIC data in the reanalyses schemes used is too low."

As is apparent in the previous discussion, I liked this paper a lot, and I think it brings out some very important points for users of stratospheric reanalyses products in their research.  Nevertheless, I do have some critical comments, which follow.

1. On page 1, line 5, COSMIC is described as an instrument. It is not. See page 3, lines 6 and 7.
2. On page 1, line 7, I suggest "use cases" be deleted, in favor of the word "usage."
3. On page 1, reference is made several times to "full input reanalyses" without definition.
4. On page 3, on line 2 the acronym COSMIC is defined incorrectly. This is done correctly lower down on line 28.
5. On page 3, line 20, "they" should be "this."
6. Page 4, line 1. Aren't stratospheric temperatures always "dry?"
7. I found the description of figure (4b) to be confusing. POerhaps, the authors might revise the text in this regard.
8. I think the paper would benefit from a more "broad-brush" description of the methodology in the main text with the details being in either an appendix or a supplement. While it is important to describe their procedures, I tended to get bogged down in the detailed procedure descriptions. This detailed description should not detract from the paper's important conclusions.
9. Page 23, lines 11-15. I find this statement confusing. The authors are attributing summer pole problems to the cold-pole problem. Isn't this a winter pole problem? Aren't the authors referring to the need for more gravity wave drag around 60 $^0$S? Certainly, this was the concern of the cited papers.
10. Would the authors say a few words describing the stand-alone black dots in figures 16 and 17?
11. Page 17, line 26, satellite should be singular.
12. Page 29, lines 12-14, is this due to the large vertical extent of the AIRS weighting function, its fine horizontal resolution, or both?
13. Page 29, line 20. The zonal mean results are very interesting. I think it might also be interesting to compare results for various wavenumbers since planetary wave diagnostics are quite common in stratospheric research.

**References**

Randel et al., 2004: The SPARC Intercomparison of Middle-Atmosphere Climatologies. *J. Climate*, *doi.org/10.1175/15200442(2004)017<0986:TSIOMC>2.0.CO;2*.

---

## Referee Comment (RC2) · S. Chabrillat (Referee) · 20 Aug 2018

**1   General comments**

This intercomparison of stratospheric temperatures in reanalyses is broad, detailed and innovative thanks to an in-depth comparison with four different satellite instruments and an interesting interpretation using the cluster analysis technique (section 10). The manuscript is very well written and the figures are excellent as long as they are seen

on a large screen: many require improvements to be readable on a print-out.

I found that Section 4 is too long and technical. It could be summarized, moving some content to the annexes (e.g. Figure 5 and most of section 4.2). The really significant information in section 4.2 is about the specification of the sensing volumes of the instruments but this is not sufficiently detailed (see specific comment below).

In my opinion this study has only one weakness: the vertical and horizontal dimensions should be separated in the comparison between full sampling and single-point sampling of AIRS observations (section 5 and appendix B). It is found that the added cost of full sampling is not justified when comparing reanalysis temperatures with HIRDLS data and seldom justified when comparing them with COSMIC and SABER data. This is an interesting (and comforting) outcome. But considering the viewing geometry of AIRS, the finding that it requires full sampling is quite trivial in the vertical dimension. For example in the case of constituent measurements, any comparison with nadir-looking instruments requires preliminary convolution of the model output by the vertical averaging kernels of the observations. Hence we are left wondering if full-sampling of AIRS in the horizontal dimension has any impact on the comparison. Of course there is no point in re-processing the whole dataset, but this question deserves either some discussion or (better) a sensitivity test. Here I would recommend to pick one year from one of the higher-resolution reanalyses and compare its fully (3D) AIRS-sampled dataset with another one where a more "usual" sampling is applied, i.e. 1D sampling in the vertical and bilinear interpolation in the horizontal.

**2  Specific comments and minor corrections**

- The abstract should explain in a few words the "full-3D sampling approach" to contrast it with the single point approximation.

- P.2, lines 1-4: You state that the biases between the reanalysis and observed

states are due not only to the multiplicity of assimilated datasets but also to the need to favour the model state for reasons of numerical stability and dynamical balance. This is a very interesting insight but it requires supporting references.

- P.2, line 29: please give a few words about the different diagnostics used in sections 6–9.

- P.3, line 20: "they preserve"

- P.3, line 21: re-phrase the sentence, e.g. "... while they are suppressed by the methods used to optimise the standard AIRS Level 2 product..."

- P.3, lines 25-26: it makes no sense to describe values derived from perturbations to synthetic data as "measurements" - please use better wording.

- P.4, line 7: Please provide specific references for the across-line-of-sight and along-line-of-sight resolutions of AIRS.

- P.4, lines 29–31 and Figures 1–2: these are very helpful and informative figures but they require some details about the methods used to approximate the sensitivity of the instruments. Note also similar question below (p.10, lines 28–32).

- Figure 1d: it is not possible to distinguish between solid and dotted lines (except looking on a screen with very high zooming)

- P.5, lines 13–14: "Each of them is widely used in the scientific community for a variety of purposes" - not yet for ERA-5 which was released very recently.

- P.5 lines 15 and 16: please define "upper-atmospheric data" in this context. Consider using the word "upper" between quotes.

- P.5 lines 20-21: it is easy to be more specific. Consider: "COSMIC is assimilated by all reanalyses except for JRA-55 and JRA-55C, AIRS by most..."

- P.5 lines 21-22: The words "Beyond these details" and "extremely" are not necessary, and Fujiwara et al. (2017) is an introductory paper - not a special issue. Consider: "The S-RIP introductory paper (Fujiwara et al., 2017) provides a detailed summary of the key features of each reanalysis".

- P.5 line 26: I think that there is a description of ERA-5 either in the ECMWF newsletter or (better) in a dedicated ECMWF technical report. Please check.

- P.10, line 24: I am not an export in satellite viewing geometries, but this really puzzles me: when vertical viewing angles are defined from instrument nadir, limb-scanning instruments should be defined as 90° - not zero !?

- P.10, lines 28–32: sensing volume parameters are an important input for this study, yet no sufficient details are given about this. Are the standard deviations in each dimension a constant for each instrument? If so, this should be written in a table. If not, what do these standard deviations depend upon? Latitude, longitude, date? Or do they differ for each observation depending on its context (e.g. surface albedo)? Please provide appropriate references for each instrument. Note also similar question above (p.4, lines 29–31).

- P. 12, line 23: please take this opportunity to define the SPA acronym (and capitalize the first letters).

- P.13, lines 10–11 and also p.32 lines 4–5: see general comment above – is this difference between SPA and full sampling due only to the vertical distribution of sensitivities or also to their horizontal distribution?

- P.13, line 12: footnote is not necessary

- P.13, line 17: delete extraneous words "and Appendix A)"

- P.13, line 18: "for COSMIC and SABER data, in particular..."

- Figure 6: it is not possible to distinguish between solid and dotted lines (except looking on a screen with very high zooming).

- Figure 6 and 7: Despite the explanation in the caption of Figure 6, the last line of text annotation (e.g. "CO=0.99C+3" or "SA=1.01E+-2") is unclear (especially for ERA-I and ERA-5 where the "E" looks like scientific notation). It would be simpler to directly write the values of gradient and the intercept separated by a comma, e.g. "(0.99,3)" or "(1.01,-2)".

- P. 14, line 6: while discussing figure 6, please remind the reader that ERA-5 is not compared with HIRDLS because you study only the post-2010 subset of ERA-5 while HIRDLS ended in 2008.

- P.15, lines 3–7: this is easy and interesting to check: are most outlying COSMIC profiles located close to the poles?

- P.18, lines 5–9: this attempt to qualitatively discuss SSW interannual variability is inadequate. Since this topic is largely out of scope, it should be sufficient to simply list the largest SSWs while dropping lines 6–8: "The SSWs of January 2006, January 2009, March 2010 and January 2012 (Butler et al., 2017) are clearly visible at both altitudes, and all datasets show a near-identical response at the 30km level. However... "

- P.19, end of section 7: Figure 10 is not discussed at all. This should be done (e.g. there is a large spread in annual cycles at 50 km) or else this figure should be dropped.

- P.19, line 14: "...acts as a 'true' estimate which the *reanalyses* are attempting to approximate."

- P.20, line 19: It may be worth mentioning that this "ability of the reanalyses to reproduce the observational record" is relatively low at 70km.

- Figures 12–15: quite difficult to visualize (especially on paper) due to the monochrome colormaps.

- Captions of Figures 12–13: re-phrase "...indicate boundary full and partial SABER coverage".

- P.23, line 14: "... resolving..."

- P.23, end of 23: on Figure 15 one also notes significantly larger RMSD in the winter polar latitudes. This should be highlighted and may be shortly discussed.

- Title of sub-section 10.2: this should not be identical to the title of section 10. For 10.2 I suggest "Co-located Cluster Analysis".

- P.29 line 19 This is still part of section 10. Replace "Section 10 suggests..." by "The previous section suggests..."

- P.26 line 31: remove extraneous ")"

- P.28 caption of Figure 18: this is an unusual graph in our field and it plays an important role in the paper, so it is important to provide a clear and complete caption (i.e. "see text for details" is insufficient). Please repeat that the co-located measurement pairs are all at 30km, horizontal distance does not imply any information, and the ordering is chosen purely to produce a simple tree.

- P.29 line 13: "... always required when comparing to AIRS,...": this may be true only w.r.t. the vertical dimension whcih would be trivial (see above)

- P.29 line 14: "... and required in equatorial regions and regions of high gravity wave activity ..." . This is only a conjecture i.e. it has not been demonstrated in this study. I suggest to tone down the conclusion: "... and may be required in ..."

- P.29 lines 27–28: as I understand them your results are not about the variability between pairs of datasets but rather about the agreement between these pairs. If this is correct, consider replacing "...variability...significantly less..." with "...agreement... significantly better..."

- P.31 lines 25–27: this is an interesting point. It should be mentioned in the body of the paper.

---

## Author Comment (AC1) · 25 Aug 2018

1. *On page 1, line 5, COSMIC is described as an instrument. It is not. See page 3, lines 6 and 7.*

We've rephrased the sentence to avoid this implication.

2. *On page 1, line 7, I suggest "use cases" be deleted, in favor of the word "usage."*

[Figure]

Done.

3. *On page 1, reference is made several times to "full input reanalyses" without definition.*

This has been clarified by adding "(those which assimilate the full suite of observations, i.e. excluding JRA-55C)" after the first mention.

4. *On page 3, on line 2 the acronym COSMIC is defined incorrectly. This is done correctly lower down on line 28.*

Fixed.

5. *On page 3, line 20, "they" should be "this."*

Fixed.

6. *Page 4, line 1. Aren't stratospheric temperatures always "dry?"*

True! We've removed the word 'dry'.

7. *I found the description of figure (4b) to be confusing. Perhaps, the authors might revise the text in this regard.*

The description has been rephrased to be clearer.

8. *I think the paper would benefit from a more "broad-brush" description of the methodology in the main text with the details being in either an appendix or a supplement. While it is important to describe their procedures, I tended to get bogged down in the*

*detailed procedure descriptions. This detailed description should not detract from the paper's important conclusions.*

We agree with this comment, which both reviewers made in some form. The current ordering arose due to the evolution of the paper: as originally planned it did not include the material after section 7 and as such was more focused on the method. We have now moved the technical details of the OIF, MIF and Core to a new Appendix, and replaced this section with a brief overview of the three components.

9. *Page 23, lines 11-15. I find this statement confusing. The authors are attributing summer pole problems to the cold-pole problem. Isn't this a winter pole problem? Aren't the authors referring to the need for more gravity wave drag around 60S? Certainly, this was the concern of the cited papers.*

You're absolutely correct - the sentence has been removed.

10. *Would the authors say a few words describing the stand-alone black dots in figures 16 and 17?*

The captions of Figures 16 and 17 have been significantly extended for clarity - including explaining the dots!

11. *Page 17, line 26, satellite should be singular.*

We believe this refers to Page 26 Line 26, which we have fixed.

12. *Page 29, lines 12-14, is this due to the large vertical extent of the AIRS weighting function, its fine horizontal resolution, or both?*

This is an important question, and one that should have been considered at time of

writing in more depth. We have now assessed this in detail, as described in our response to the other comment on this article by Simon Chabrillat.

Specifically, we find that while a 1D (i.e. considering the vertical form only) approach does achieve most of the gains of the 3D approach, more than 50% of samples retain an above-instrument-precision difference, of up to 5 K. This has been added to the text both in the main body and as an additional section of the relevant appendix.

13. *Page 29, line 20. The zonal mean results are very interesting. I think it might also be interesting to compare results for various wavenumbers since planetary wave diagnostics are quite common in stratospheric research.*

We also agree this would be interesting. We are currently in the data-analysis stage of a follow-up paper focusing on gravity waves as-resolved by these reanalyses, and now plan to include planetary wave diagnostics in response to this comment.

---

## Author Comment (AC2) · 25 Aug 2018

**1. General comments**

*the figures are excellent as long as they are seen on a large screen: many require improvements to be readable on a print-out.*

The reviewer provides specific comment about figures 1d, 6, and 12-15 below. We have addressed these individually, and discuss the changes made in the specific comments section.

[Figure]

*I found that Section 4 is too long and technical. It could be summarized, moving some content to the annexes (e.g. Figure 5 and most of section 4.2). The really significant information in section 4.2 is about the specification of the sensing volumes of the instruments but this is not sufficiently detailed (see specific comment below).*

We agree with this comment, which both reviewers made in some form. The current ordering arose due to the evolution of the paper: as originally planned it did not include the material after section 7 and as such was more focused on the method. We have now moved the technical details of the OIF, MIF and Core to a new Appendix, and replaced this section with a brief overview of the three components.

*In my opinion this study has only one weakness: the vertical and horizontal dimensions should be separated in the comparison between full sampling and single-point sampling of AIRS observations (section 5 and appendix B). It is found that the added cost of full sampling is not justified when comparing reanalysis temperatures with HIRDLS data and seldom justified when comparing them with COSMIC and SABER data. This is an interesting (and comforting) outcome. But considering the viewing geometry of AIRS, the finding that it requires full sampling is quite trivial in the vertical dimension. For example in the case of constituent measurements, any comparison with nadir-looking instruments requires preliminary convolution of the model output by the vertical averaging kernels of the observations. Hence we are left wondering if full-sampling of AIRS in the horizontal dimension has any impact on the comparison. Of course there is no point in re-processing the whole dataset, but this question deserves either some discussion or (better) a sensitivity test. Here I would recommend to pick one year from one of the higher-resolution reanalyses and compare its fully (3D) AIRS-sampled dataset with another one where a more "usual" sampling is applied, i.e. 1D sampling in the vertical and bilinear interpolation in the horizontal.*

This is an important question, and one that should have been considered in more detail in the original study. To properly assess this, we have re-run our AIRS sampling routine, using settings intended to simulate a 1D-only AIRS sampling while still allowing us to re-use the same software and thus to not introduce new inconsistencies. Specifically, we have re-run the AIRS analysis as described in the original text, except with no rotation in the horizontal or vertical and with horizontal weighting functions of width ~100 m in both directions. Since the Core routine always centres the sampled points at the centre of the measurement volume and the grid spacing required for multiple points is of order kilometres, this results in a single column of points for each sample calculation, which are then summed using the same vertical weighting functions used for the original 3D analysis to produce equivalent synthetic measurements. We have then run this over all reanalyses for the year 2011.

We find that while a 1D approach does achieve most of the gains of the 3D approach, more than 50% of samples retain an above-instrument-precision difference, of up to 5 K. This has been added to the text both in the main body and as an additional section of the relevant appendix.

**2 Specific comments and minor corrections**

*1. The abstract should explain in a few words the "full-3D sampling approach" to contrast it with the single point approximation.*

The phrase "(i.e. one which takes full account of the instrument measuring volume)" has been added at the first mention of full-3D sampling.

*P.2, lines 1-4: You state that the biases between the reanalysis and observed states are due not only to the multiplicity of assimilated datasets but also to the need to favour the model state for reasons of numerical stability and dynamical balance. This is a very interesting insight but it requires supporting references.*

A relevant citation has been added.

*P.2, line 29: please give a few words about the different diagnostics used in sections 6–9.*

Done!

*P.3, line 20: "they preserve"*

'They' has been replaced with 'this', which fixes this problem.

*P.3, line 21: re-phrase the sentence, e.g. "... while they are suppressed by the methods used to optimise the standard AIRS Level 2 product..."*

The above change should also resolve this issue.

*P.3, lines 25-26: it makes no sense to describe values derived from perturbations to synthetic data as "measurements" - please use better wording.*

The sentence has been removed, as it doesn't add to the paper anyway.

*P.4, line 7: Please provide specific references for the across-line-of-sight and along-line-of-sight resolutions of AIRS.*

We believe this refers to COSMIC, for which we have added a suitable reference (Hindley et al, ACP 2015)

*P.4, lines 29–31 and Figures 1–2: these are very helpful and informative figures but they require some details about the methods used to approximate the sensitivity of the*

*instruments. Note also similar question below (p.10, lines 28–32).*

See response below.

*Figure 1d: it is not possible to distinguish between solid and dotted lines (except looking on a screen with very high zooming)*

We have replaced this with a solid line, and modified the caption accordingly.

*P.5, lines 13–14: "Each of them is widely used in the scientific community for a variety of purposes" - not yet for ERA-5 which was released very recently.*

Clarified.

*P.5 lines 15 and 16: please define "upper-atmospheric data" in this context. Consider using the word "upper" between quotes.*

We have added the phase "(in this context, stratospheric and mesospheric)" to clarify our meaning.

*P.5 lines 20-21: it is easy to be more specific. Consider: "COSMIC is assimilated by all reanalyses except for JRA-55 and JRA-55C, AIRS by most..."*

Rephrased as suggested.

*P.5 lines 21-22: The words "Beyond these details" and "extremely" are not necessary, and Fujiwara et al. (2017) is an introductory paper - not a special issue. Consider: "The S-RIP introductory paper (Fujiwara et al., 2017) provides a detailed summary of the key features of each reanalysis".*

Rephrased as suggested.

*P.5 line 26: I think that there is a description of ERA-5 either in the ECMWF newsletter or (better) in a dedicated ECMWF technical report. Please check.*

We have added a reference to ECMWF Newletter 147, which summarises key information about ERA-5.

*P.10, line 24: I am not an export in satellite viewing geometries, but this really puzzles me: when vertical viewing angles are defined from instrument nadir, limb scanning instruments should be defined as 90deg - not zero !?*

This has been changed to 90 degrees. Our code specified a value of zero degrees as described in the original text, but also had the horizontal and vertical volume-width parameters for each limb sounder specified the wrong way around, which cancelled out the conceptual error in angle to produce the same final averaging volume (hence why we didn't spot the error in tests!).

*P.10, lines 28–32: sensing volume parameters are an important input for this study, yet no sufficient details are given about this. Are the standard deviations in each dimension a constant for each instrument? If so, this should be written in a table. If not, what do these standard deviations depend upon? Latitude, longitude, date? Or do they differ for each observation depending on its context (e.g. surface albedo)? Please provide appropriate references for each instrument. Note also similar question above (p.4, lines 29–31).*

A paragraph has been added to the definition of the OIF explaining the assumptions implicit in this specification.
*P. 12, line 23: please take this opportunity to define the SPA acronym (and capitalize the first letters).*

Done!

*P.13, lines 10–11 and also p.32 lines 4–5: see general comment above – is this difference between SPA and full sampling due only to the vertical distribution of sensitivities or also to their horizontal distribution?*

See discussion in 'General comments' section, above.

*P.13, line 12: footnote is not necessary*

Removed.

*P.13, line 17: delete extraneous words "and Appendix A)"*

Removed.

*P.13, line 18: "for COSMIC and SABER data, in particular..."*

Fixed.

*Figure 6: it is not possible to distinguish between solid and dotted lines (except looking on a screen with very high zooming).*

The caption has been modified to make clear that it essentially overlies the solid line.

*Figure 6 and 7: Despite the explanation in the caption of Figure 6, the last line of text annotation (e.g. "CO=0.99C+3" or "SA=1.01E+-2") is unclear (especially for ERA-I*

*and ERA-5 where the "E" looks like scientific notation). It would be simpler to directly write the values of gradient and the intercept separated by a comma, e.g. "(0.99,3)" or "(1.01,-2)".*

The annotations have been replaced with the form y=mx+c - this provides a compromise between the two forms and is clearer than the original.

*P. 14, line 6: while discussing figure 6, please remind the reader that ERA-5 is not compared with HIRDLS because you study only the post-2010 subset of ERA-5 while HIRDLS ended in 2008.*

A footnote has been added clarifying this.

*P.15, lines 3–7: this is easy and interesting to check: are most outlying COSMIC profiles located close to the poles?*

In fact, the data we present later in the paper already tests this, although we had not made the mental link! A cross-reference has now been added to the later section on geographically-localised comparisons, where we see that at 30 km altitude this does not appear to be a significant factor.

*P.18, lines 5–9: this attempt to qualitatively discuss SSW interannual variability is inadequate. Since this topic is largely out of scope, it should be sufficient to simply list the largest SSWs while dropping lines 6–8: "The SSWs of January 2006, January 2009, March 2010 and January 2012 (Butler et al., 2017) are clearly visible at both altitudes, and all datasets show a near-identical response at the 30km level. However... "*

Changed as suggested.

*P.19, end of section 7: Figure 10 is not discussed at all. This should be done (e.g. there*

*is a large spread in annual cycles at 50 km) or else this figure should be dropped.*

A couple of sentences have been added describing this figure in more detail.

*P.19, line 14: "...acts as a 'true' estimate which the \*reanalyses\* are attempting to approximate."*

Fixed.

*P.20, line 19: It may be worth mentioning that this "ability of the reanalyses to reproduce the observational record" is relatively low at 70km.*

We have added the clause "however, it is also low in absolute terms due to the difficulties of modelling this atmospheric region and the limited data constraints available." to clarify this.

*Figures 12–15: quite difficult to visualize (especially on paper) due to the monochrome colormaps.*

This was an attempt to use a colourscale based on the SRIP multi-model-mean colour, which is a fairly murky brown. However, we agree these figures are hard to read, and have replaced them with a diverging red-yellow-blue colour table to emphasise the features better.

*Captions of Figures 12–13: re-phrase "...indicate boundary full and partial SABER coverage".*

Added the word 'of' to fix the sentence.

*P.23, line 14: "... resolving..."*
Fixed.

*P.23, end of 23: on Figure 15 one also notes significantly larger RMSD in the winter polar latitudes. This should be highlighted and may be shortly discussed.*

Mentioned and some discussion added.

*Title of sub-section 10.2: this should not be identical to the title of section 10. For 10.2 I suggest "Co-located Cluster Analysis".*

Changed as suggested.

*P.29 line 19 This is still part of section 10. Replace "Section 10 suggests..." by "The previous section suggests..."*

Changed as suggested - this was a hangover from an earlier version with separate sections for these topics.

*P.26 line 31: remove extraneous ")"*

Fixed.

*P.28 caption of Figure 18: this is an unusual graph in our field and it plays an important role in the paper, so it is important to provide a clear and complete caption (i.e. "see text for details" is insufficient). Please repeat that the co-located measurement pairs are all at 30km, horizontal distance does not imply any information, and the ordering is chosen purely to produce a simple tree.*

A more detailed description has been added to the caption, including all the requested points.

[Figure]

*P.29 line 13: "... always required when comparing to AIRS,...": this may be true only
w.r.t. the vertical dimension whcih would be trivial (see above)*

See discussion in 'General comments' section, above.

*P.29 line 14: "... and required in equatorial regions and regions of high gravity wave
activity ..." . This is only a conjecture i.e. it has not been demonstrated in this study. I
suggest to tone down the conclusion: "... and may be required in ..."*

Changed as suggested.

*P.29 lines 27–28: as I understand them your results are not about the variability be-
tween pairs of datasets but rather about the agreement between these pairs. If this is
correct, consider replacing "...variability...significantly less..." with "...agreement... sig-
nificantly better..."*

Agreed and changed.